# Latent Noise Segmentation: How Neural Noise Leads to the Emergence of Segmentation and Grouping

## Abstract

Deep Neural Networks (DNNs) that achieve human-level performance in general tasks like object segmentation typically require supervised labels. In contrast, humans are able to perform these tasks effortlessly without supervision. To accomplish this, the human visual system makes use of perceptual grouping: for example, the black and white stripes of a zebra are perceptually grouped together despite their vastly different colors. Understanding how perceptual grouping arises in an unsupervised manner is critical for improving both computer vision models and models of the visual system. In this work, we propose a counterintuitive approach to unsupervised perceptual grouping and segmentation: that they arise *because* of neural noise, rather than in spite of it. We (1) mathematically demonstrate that under realistic assumptions, neural noise can be used to separate objects from each other, and (2) show that adding noise in a DNN enables the network to segment images even though it was never trained on any segmentation labels. Interestingly, we find that (3) segmenting objects using noise results in segmentation performance that aligns with the perceptual grouping phenomena observed in humans. We introduce the Good Gestalt (GG) datasets — six datasets designed to specifically test perceptual grouping, and show that our DNN models reproduce many important phenomena in human perception, such as illusory contours, closure, continuity, proximity, and occlusion. Finally, we (4) demonstrate the ecological plausibility of the method by analyzing the sensitivity of the DNN to different magnitudes of noise. We find that some model variants consistently succeed with remarkably low levels of neural noise ($\sigma < 0.001$), and surprisingly, that segmenting this way requires as few as a handful of samples. Together, our results suggest a novel unsupervised segmentation method requiring few assumptions, a new explanation for the formation of perceptual grouping, and a novel potential benefit of neural noise.

## 1 Introduction and Related Work

Humans perceptually group object parts in scenes (Wagemans et al., 2012a;b; Jäkel et al., 2016; Roelfsema, 2006), which allows them to interpret and segment elements in context, even when they deviate from stored templates (Geirhos et al., 2019; Herzog et al., 2015; Holcombe and Cavanagh, 2001). Historically, perceptual grouping in humans has been studied by the Gestaltists, who view the Gestalt principles of grouping (such as grouping by proximity and continuity) as crucial computational principles. This human capability to perform perceptual grouping is in stark contrast to object recognition in Deep Neural Networks (DNNs), which have been shown to struggle with both robustness (Geirhos et al., 2018; Madry et al., 2018; Moosavi-Dezfooli et al., 2016) and grouping (Bowers et al., 2022; Linsley et al., 2018; Doerig et al., 2020; Biscione and Bowers, 2023). Understanding the underlying mechanisms of perceptual grouping is critical for better modeling of human vision, and a more fundamental understanding of segmentation in general.

While incorporating explicit or implicit segmentation and grouping mechanisms in DNNs has seen substantial rise in interest in recent years (Greff et al., 2020; Hinton, 2023; Linsley et al., 2018; Sabour et al., 2017), mainstream approaches often either train models directly to segment using labeled samples (Ronneberger et al., 2015; Long et al., 2015; Kirillov et al., 2023), or inject a sub-

stantial number of biases into the model architecture (Hamilton et al., 2022; Engelcke et al., 2022; 2020a). The current prevailing approach is the implementation of object slots in the model architecture (Locatello et al., 2020), which maintains information about object instances in a separately maintained vector. Slot-based approaches are conceptually limited by the fact that the number of slots is fixed in the model architecture and cannot be adapted by learning, nor on an image-by-image basis. An approach that attempts to remedy the problems above was proposed by Löwe et al. (2022), who suggest a neuroscience-inspired solution to object learning: the object identities are stored in phase values of complex variables, which mimics the temporal synchrony hypothesis of brain function (Milner, 1974). While the approach can handle a variable number of objects, it is limited in the maximum number of objects it can represent (but see Stanić et al. (2023)). To date, there is no unified and simple principle that allows DNNs to perform segmentation and grouping in a number of different, seemingly unrelated contexts with few *in-principle* limitations. In this work, we seek to remedy this gap.

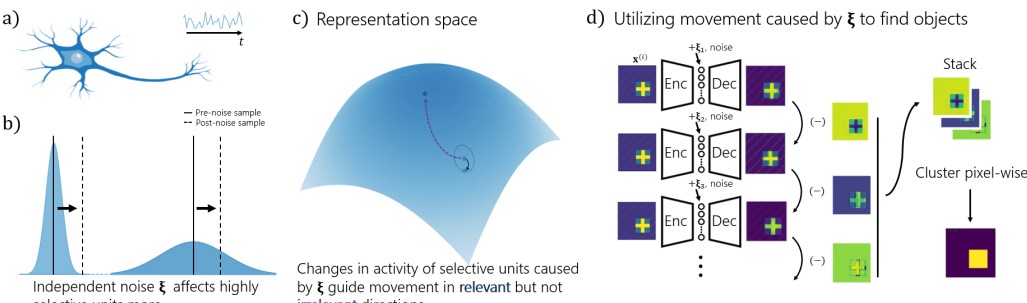

Figure 1: **Latent Noise Segmentation Schematic Illustration.** (a) Biological neurons are highly noisy. For example, thermal noise and ion channel shot noise (Manwani and Koch, 1998) contribute to independent noise in neurons. (b) Independent noise affects the output of neurons that are highly selective to a stimulus feature (*left*) than neurons that are less selective (*right*). Solid lines indicate the mean of the noise-free activity distribution, and dashed lines indicate the actual sample after independent noise is added. *The x-axis in the illustration is not meaningful.* (c) In the system's representational space (as indicated by the surface where input images are mapped to points on that surface), the changes caused by independent noise cause meaningful changes to the model's representation in relevant directions (e.g., local PC directions), but not irrelevant directions (that would substantially change the model's representation of the input). (d) This yields information about the objects in the input image, and can be used to segment the input images. An input image $\mathbf{x}^{(i)}$ is fed into an autoencoder network, and noisy samples are drawn and consecutively subtracted from each other. These outputs contain information about the changes induced by noise in latent space, cast into image space. The outputs are stacked and clustered pixel-wise to generate a segmentation mask.

Specifically, we develop a direct test benchmark of a network's segmentation and grouping capability by requiring it to segment images in a way that necessitates perceptual grouping —- furthermore, the network must do this without ever being trained on the stimuli in question, and without ever being trained on the task of segmentation, similarly to how the human visual system is not. To simultaneously accomplish all these goals, we posit a seemingly counterintuitive idea that we call Latent Noise Segmentation (LNS):

*Neural noise enables deep neural networks to perceptually group and segment.*

See Figure 1 for a schematic overview of the approach. In brief, we show that by injecting independent noise to a hidden layer of a DNN, we can turn DNNs trained on generic image reconstruction tasks to systems capable of perceptual grouping and segmentation.

In the following sections, we describe the general setting of LNS and its specific implementation in an autoencoder DNN (Section 2). We mathematically demonstrate that under certain (relatively loose) assumptions, the presence of latent noise allows visual entities, which do not perfectly covary with each other, to be separated into individual objects (Section 2.1 and Appendix A.1). We develop a comprehensive test benchmark of segmentation tasks inspired by a century of Gestalt psy-

chology and the study of grouping in humans (Wertheimer, 1923; Koffka, 1922) that we call the Good Gestalt (GG) datasets. We empirically show that LNS directly reproduces the result of the elusive Gestalt principles of grouping in segmentation masks (Section 2.3). Finally, we study how practically feasible LNS is (that is, how in-principle usable the method is by any system constrained by limitations like compute time or noise magnitude, such as the primate visual system) by evaluating how segmentation performance varies with different model learning rules, noise levels, and the number of time steps the model takes to segment (Section 3). Our results suggest that a practically feasible number of time steps (as few as a handful) are sufficient to reliably segment and that while encouraging a prior distribution in a model does not improve its segmentation performance, it stabilizes the optimal amount of noise needed for segmentation across all datasets and correctly identifies the appropriate number of objects.

## 2 MODEL AND METHODS

Here, we describe the general setting of our methodology, how we obtain segmentation masks with LNS, the design of our datasets, and model evaluation metrics.

**Basic setup.** Our model architecture consists of two primary components. The backbone of the model is a pre-trained (Variational) Autoencoder (VAE; Kramer (1991); Kingma and Welling (2014); Higgins et al. (2017)), where an encoder $\text{Enc}(\mathbf{x}) = q_\phi(\mathbf{z}|\mathbf{x})$ learns a compressed latent distribution, and a decoder $\text{Dec}(\mathbf{z}) = p_\theta(\mathbf{x}|\mathbf{z})$ models the data that generated the latent representation $\mathbf{z}$. In practice, the model is optimized using the Evidence Lower Bound (ELBO):

$$ELBO = \mathbb{E}_{q_\phi(\mathbf{z}|\mathbf{x})}[\log p_\theta(\mathbf{x}|\mathbf{z})] - \beta D_{KL}(q_\phi(\mathbf{z}|\mathbf{x})||p(\mathbf{z})), \tag{1}$$

where $D_{KL}$ is the Kullback-Leibler divergence, and $p(\mathbf{z})$ is the prior set to $\mathcal{N}(0, \mathbf{I})$. $\beta$ is a configurable coefficient facilitating reconstruction quality, which we set automatically through the GECO loss (Rezende and Viola, 2018). For a discussion about alternate architectures, see Appendix A.1.5.

---

**Algorithm 1** Latent Noise Segmentation.

---

**Require:** image $\mathbf{x}^i$, time steps $N$, small noise variance $\sigma_{small}^2$
**Require:** trained $(\beta)$-VAE with $\text{Enc}_\phi, \text{Dec}_\theta$
 1: **procedure** SEGMENT($\mathbf{x}^i, N$)
 2: $\quad \boldsymbol{\mu}(\mathbf{x}^i) \leftarrow \text{Enc}(\mathbf{x}^i)$ $\qquad\qquad\qquad\qquad\qquad\qquad\qquad$ ▷ Get latent unit means
 3: $\quad$ **for** $n = 1, \dots, N$ **do**
 4: $\qquad \boldsymbol{\xi}_n \leftarrow \mathcal{N}(0, \sigma_{small}^2)$ $\qquad\qquad\qquad\qquad\qquad\qquad$ ▷ i.i.d. Gaussian noise
 5: $\qquad \tilde{\mathbf{x}}_n^i \leftarrow \text{Dec}\left(\boldsymbol{\mu}\left(\mathbf{x}^i\right) + \boldsymbol{\xi}_n\right)$ $\qquad\qquad\qquad$ ▷ Save perturbed decoder output
 6: $\qquad$ **if** $n \geq 2$ **then**
 7: $\qquad\quad \Delta\tilde{\mathbf{x}}_{n-1}^i \leftarrow \tilde{\mathbf{x}}_n^i - \tilde{\mathbf{x}}_{n-1}^i$ $\qquad\qquad\qquad\qquad$ ▷ Subtract outputs
 8: $\qquad$ **end if**
 9: $\quad$ **end for**
10: $\quad$ Let $\Delta\tilde{\mathbf{X}}^i \equiv [\Delta\tilde{\mathbf{x}}_1^i, \Delta\tilde{\mathbf{x}}_2^i, \dots, \Delta\tilde{\mathbf{x}}_{N-1}^i]$ ▷ $\Delta\tilde{\mathbf{X}}^i$ has dimension $img_x \times img_y \times c \times (N-1)$
11: $\quad$ Cluster $\Delta\tilde{\mathbf{X}}^i$ $\qquad\qquad\qquad\qquad\qquad\qquad\qquad$ ▷ Separate pixel identities
12: $\quad$ Assign segmentation mask values to pixels according to clustering
13: **end procedure**

---

### 2.1 LATENT NOISE SEGMENTATION

After training the model backbone, we build a noisy segmentation process which enables the extraction and combination of information from the model's latent space to generate segmentation masks. To do this, we add i.i.d. noise $\boldsymbol{\xi}_n$ to the latent variables $\mathbf{z}$. The process is repeated $N$ times, allowing us to obtain information about which objects exist in the model outputs. For every time step, two consecutive model outputs are subtracted from each other to find what in the output images $\mathbf{x}$ *changed with respect to changes in latent space* $\mathbf{z}$. At the end of the process, the stack of subtracted outputs is clustered using Agglomerative Clustering (Pedregosa et al., 2011), resulting in a segmentation mask. Full details of the algorithm are shown in Algorithm 1, a schematic is shown in Figure 1, and additional details can be found in Appendix E.2.

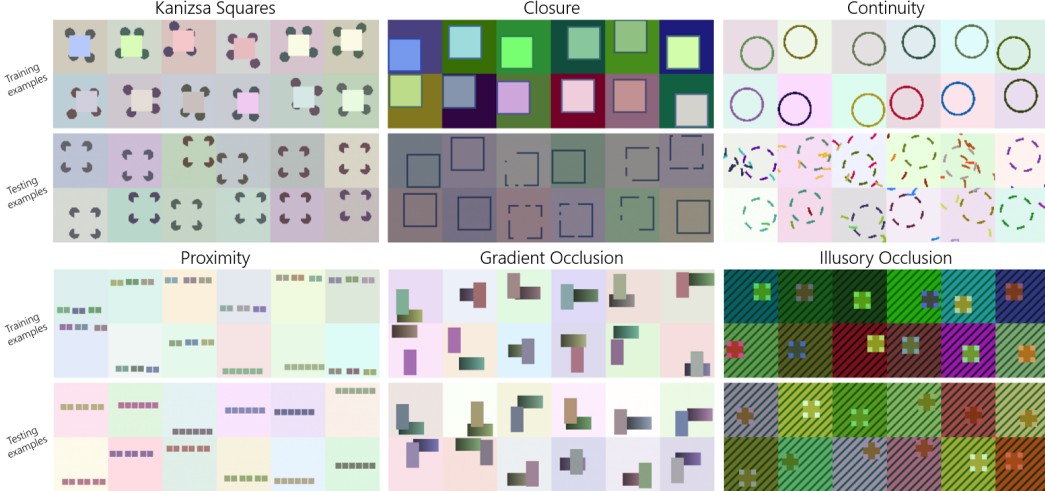

Figure 2: **The Good Gestalt (GG) Datasets.** Zoom in for an ideal viewing experience. The first two rows of images show training image examples, while the second two rows show images of testing examples.

## 2.2 THE ROLE OF INDEPENDENT NOISE

Latent Noise Segmentation works due to several key reasons. Independent noise $\boldsymbol{\xi}$ induces the locally monotone mapping $f : \Delta\mathbf{z} \rightarrow \Delta\mathbf{x}$ learned by the model. A potential problem is that this mapping happens for all units — even those that are not meaningful for the task of segmentation — and as such, they might adversely affect segmentation performance. To solve this, one could identify the meaningful units by perturbing them individually (like in latent traversal (Kingma and Welling, 2014; Higgins et al., 2017), but not only is this computationally expensive and impractical, it is also biologically implausible.

The solution to the problem comes from the intuition that units that meaningfully code for relevant stimulus features have a substantially higher derivative $\Delta\mathbf{x} = \Delta\mathbf{z} * \frac{\delta\mathbf{x}}{\delta\mathbf{z}}$ in the neighborhood of the stimulus $\mathbf{x}$. This is because for the decoder to achieve its goal of low reconstruction loss, it should be sensitive to small changes in the activity of units that are coding for relevant features in $\mathbf{x}$, but not for irrelevant features. This means that we can perturb all units simultaneously with independent noise: training on a generic reconstruction loss encourages the network to code in a manner that allows independent noise to reveal the relevant derivative direction, making the representation clusterable for segmentation (Figure 1b). Indeed, the goal of independent noise is not to perform segmentation itself, but to reveal the local neighborhood of the input stimulus and thus enable segmentation, *from the perspective of the decoder*.

We provide a more complete mathematical intuition of exactly why noise perturbation on latent units causes a meaningful object segmentation in the case of the VAE in Appendix A.1. In summary, we observe that noise induces the output pixels belonging to the same part of the image to co-vary with each other, while pixels belonging to different parts do not. Following a principal component argument (Zietlow et al., 2021; Rolínek et al., 2019), we show that the derivatives of the decoder's output with respect to the latent units are the local principal component directions of the training data. Together, these phenomena enable meaningful object segmentation.

## 2.3 THE GOOD GESTALT (GG) DATASETS

To evaluate our models, we developed the six datasets that we together call the Good Gestalt (GG) datasets. The notion of Gestalt comes from the study of perception (Wertheimer, 1923; Jäkel et al., 2016; Wagemans et al., 2012a;b), and is perhaps best summarized as 'the whole is greater than the sum of its parts'. The GG datasets aim to address a large variety of different rules of perceptual organization, otherwise known as Gestalt principles of grouping like continuity, closure and prox-

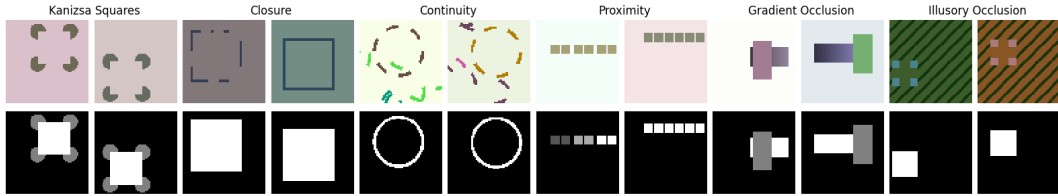

Figure 3: **Target segmentation mask examples of the GG datasets.** The first row of images shows training image examples, while the second row shows images of testing examples. Different colors indicate different objects. **Kanizsa Squares**: The model should segment a square in the center, as well as four background circles, separately from the background. **Closure**: The model should segment a square. **Continuity**: The model should segment a circle traced by the relevant line segments; to "complete the circle". **Proximity**: The model should segment a set of six squares together, or three sets of two squares when the proximity cue is given. **Gradient Occlusion**: The model should segment the two rectangles separately. **Illusory Occlusion**: The model should segment the static background and stripes together, and the foreground object parts together.

imity — here, we outline the reasons for studying machine perception using such datasets, and then describe the specifics of our datasets. Example stimuli of the training and testing samples are shown in Figure 2.

**Reasons for studying Gestalt.** In almost all the Gestalt stimuli we study in this work, we assume that in training the luminances (pixel values) of different objects in a visual scene are independent, while the luminances (pixel values) of the same object co-vary. This assumption follows a simplified view on optics (for details, see Appendix E.1), and is important as some principles of perceptual grouping have been shown to be consistent with the statistical structure of the natural environment (Elder and Goldberg, 2002; Geisler et al., 2001; Sigman et al., 2001). In testing, the model should generalize this information in ambiguous contexts, where the pixel values of the images are not fully informative of the object. For example, in the Kanizsa Squares dataset (Figure 2, *Kanizsa Squares, Testing examples*), the pixel value of the background is the same as the pixel value of the illusory square in the middle of the cornering circles.

We show target segmentation masks for the GG datasets in Figure 3. To determine what these segmentation masks should be, our focus here is not about *what* specifically humans perceive in a specific psychophysical experiment, but rather about what humans *can* perceive given the right instruction. This is part of what makes studying Gestalt so difficult: simple phrasing about what a participant should segment in such an image can potentially crucially affect how they segment the image. This is why in our GG datasets, we did not collect human data, but instead focus on the generic question of "which elements in the image belong together?", and defined the target segmentation masks for the test set evaluation on the basis of this question, focusing on evidence from human research in general from the past century (Wertheimer, 1923; Kimchi, 1992).

In total, the GG datasets are made up of six individual datasets, for which we explain the main premise individually below. More information on the datasets can be found in Appendix E.1.

**Kanizsa Squares** to study the perception of illusory figures. Humans are able to perceive illusory contours or illusory objects when cued by, for example, Pacman-like circles (Wang et al., 2012; Lesher, 1995).

**Closure** to study whether a model can combine individual image elements to form an object. The human ability to group objects together when they form a closed figure is well-documented (Elder and Zucker, 1993; Pomerantz et al., 1977; Ringach and Shapley, 1996; Marini and Marzi, 2016).

**Continuity** to study the integration of individual elements across the visual field. When elements are oriented such that they would form a continuous object, humans are able to group the elements together to form a whole figure (Kwon et al., 2016; Kovács and Julesz, 1993; Field et al., 1993).

**Proximity** to study the cue of intra-element distance for grouping. When multiple elements in a figure are similar, humans can perceive many similar elements as belonging together with

those that are closer in proximity (Kubovy and van den Berg, 2008; Quinlan and Wilton, 1998).

**Gradient Occlusion** to study the ability to handle full or partial occlusion of an element without a constant color value. The basis of objecthood in humans does not depend crucially on the exact color of the elements, and long-range integration and inference about objects and how they change under occlusion is possible (Kellman and Spelke, 1983; Spelke, 1990).

**Illusory Occlusion** to study grouping across the visual field while being occluded by an illusory object. Regular textures can be grouped together, and foreground objects that break the pattern can be separated from the complex textural background. A classic example is the Dalmatian Dog illusion by R. C. James, of which our Illusory Occlusion dataset is a simplification.

Together, the GG datasets comprise tests of a substantial number of the most major Gestalt laws of perceptual organization (for reviews, see Wagemans et al. (2012a;b); Jäkel et al. (2016); Todorovic (2008); Kanizsa (1979)), including figure-ground segregation, proximity, continuity, closure, and the general notion of Good Gestalt (the tendency of the perceptual system to organize objects according to the aforementioned laws).

## 2.4 EVALUATION METRICS AND BASELINE MODELS

**Metrics.** We evaluate all models using the Adjusted Rand Index (ARI) metric (Hubert and Arabie, 1985), which measures the above-chance similarity between two clusterings: the target segmentation mask, and the predicted segmentation mask. Here we choose to judge our models based on the ARI metric (instead of ARI minus background) because the point of the GG datasets (and of the study of Gestalt in general) is that of object-background separation, and not of fine-grained object-specific details. We verified that in all of the GG datasets, an all-background assignment yields an ARI of $\approx 0$, i.e. no better than random assignment. In addition, we plot randomly drawn examples of segmentation masks in Figure 4, and all segmentation masks for our highest-performing models in Appendix F.4, Figures 16 - 27.

**Models and control experiments.** We trained 5 Autoencoder (AE) and Variational Autoencoder (VAE) models with MSE and GECO (Rezende and Viola, 2018) losses, respectively. In addition, we computed two control experiments, one for each model variant (Noisy AE/VAE Reconstruction) with the goal of verifying whether adding noise at the latent layer is important. These control experiments compute a clustering on the AE/VAE reconstruction pixel values, adding noise to the reconstruction instead of the latent layer, keeping the rest of the model architecture and methodology the same. Implementation details and model architectures are shown in Appendix E.2, and reconstructions are shown in Appendix 14. We also evaluate two baseline models, Genesis (Engelcke et al., 2020a) and Genesis-v2 (Engelcke et al., 2022), for which details are shown in Appendix C.

## 3 EXPERIMENTAL RESULTS

### 3.1 QUANTITATIVE EVALUATION

| Model | Kanizsa | Closure | Continuity | Proximity | Gradient Occ. | Illusory Occ. |
|---|---|---|---|---|---|---|
| AE | $0.859 _{\pm 0.008}$ | $0.766 _{\pm 0.008}$ | $0.552 _{\pm 0.008}$ | $\mathbf{0.996^*} _{\pm 0.001}$ | $0.926 _{\pm 0.009}$ | $\mathbf{0.994^*} _{\pm 0.002}$ |
| VAE | $\mathbf{0.871} _{\pm 0.005}$ | $\mathbf{0.795} _{\pm 0.009}$ | $0.593 _{\pm 0.012}$ | $0.943 _{\pm 0.010}$ | $0.918 _{\pm 0.002}$ | $0.974 _{\pm 0.003}$ |
| AE Rec. | $0.246 _{\pm 0.006}$ | $0.064 _{\pm 0.001}$ | $0.570 _{\pm 0.006}$ | $0.141 _{\pm 0.001}$ | $\mathbf{0.982} _{\pm 0.000}$ | $-0.023 _{\pm 0.001}$ |
| VAE Rec. | $0.343 _{\pm 0.004}$ | $0.084 _{\pm 0.000}$ | $\mathbf{0.611} _{\pm 0.005}$ | $0.144 _{\pm 0.001}$ | $0.977 _{\pm 0.001}$ | $-0.024 _{\pm 0.001}$ |
| Genesis | 0.346 | 0.755 | 0.394 | 0.879 | 0.933 | 0.294 |
| Genesis-v2 | 0.415 | 0.000 | 0.399 | 0.059 | 0.000 | 0.422 |

Table 1: Model results (ARI $\pm$ standard error of the mean). AE (Autoencoder) and VAE (Variational Autoencoder) model scores reported for the best noise level and number of samples. AE Rec. and VAE Rec. stand for noisy reconstruction control experiments, and (see Section 2.4 for details). Bold indicates top-performing model. For the AE and the VAE, a star (*) indicates a statistically significant difference in model performance (Bonferroni corrected).

Both the AE and VAE models achieve good quantitative performance in general perhaps with the exception of the Continuity dataset (Table 2). Adding noise in latent space as opposed to image space unlocks a substantial improvement in segmentation performance, improving ARI by 0.506 on average across all GG datasets. As described in Section 3.2, we do not see the somewhat lower quantitative score as a failure *per se*, but rather as related to the way in which the models fundamentally segment using LNS. Since the segmentation process seeks to vary the latent representation of the models, the segmentation process introduces artifacts related to the model's representational space, such as small shifts of the stimulus in a given direction, or small changes to its size. The ARI metric is punishing to even single-pixel shifts of the target object — qualitatively, it is clear that even in cases where the model does not achieve an excellent ARI score, it captures the desired effect. This is in contrast to the Genesis and Genesis-v2 models, which fail at capturing the desired effect even when they do well quantitatively C. A similar effect of minor performance deterioration with respect to the baseline of clustering a reconstruction can be observed in the Gradient Occlusion dataset, where good reconstruction quality of the model is congruent with a good segmentation performance under output clustering. This is to be expected - the dataset by design is such that the pixel values are mostly informative of the correct segmentation label identities.

To understand whether a structured latent prior could improve model segmentation capability under LNS, we conducted a two-sample two-tailed t-test to compare the AE and VAE ($\beta > 1$) performance. The test indicated that the AE and VAE performance statistically significantly differed only for the Proximity ($t = 4.54$, $p < 0.05$) and Illusory Occlusion datasets ($t = 5.22$, $p < 0.05$) after the Bonferroni correction, while no statistically significant difference was found for the other datasets. Due to the additional loss term that places tension on the quality of the reconstruction, it is not surprising that the VAE would show small decreases in performance — indeed, we conclude that encouraging a structured latent prior on the model does not in general improve the model's capability to segment stimuli.

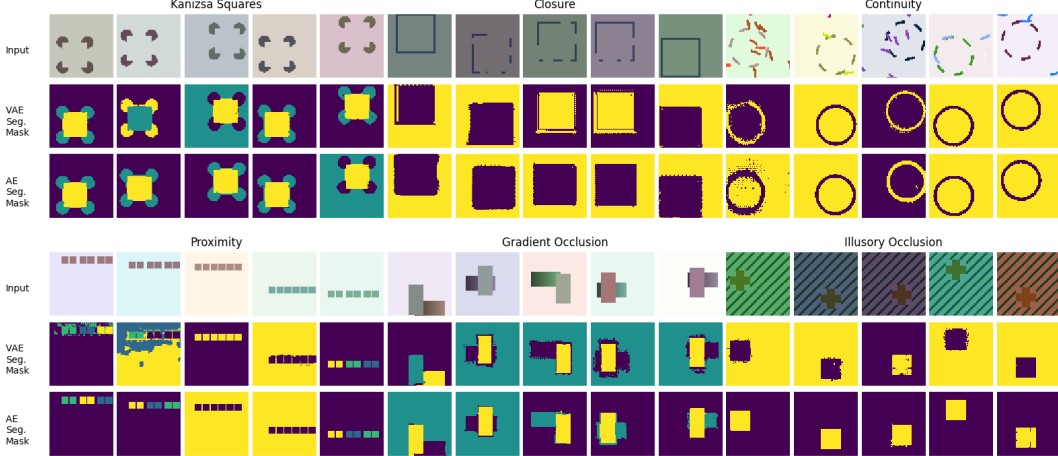

Figure 4: **Model Output Segmentation Mask Examples.** The first row shows inputs, the second row shows VAE segmentation masks, and the third row shows AE segmentation masks. Randomly selected examples from the **Kanizsa Squares**, **Closure**, **Continuity**, **Proximity**, **Gradient Occlusion**, and **Illusory Occlusion** datasets are shown using the best model hyperparameters shown in Table 2. Since the model segments by clustering noisy outputs, the specific color assignment to different object identities is arbitrary (for example, whether the model assigns the identity represented by yellow as the background, or purple, is not a meaningful distinction).

## 3.2 QUALITATIVE EVALUATION

To understand how the models perform qualitatively, we plot a random selection of examples in Figure 4, and all model outputs of the shown seed for all datasets in Appendix F.4. Important are two things: first, while model outputs are consistently good and generally reflect appropriate grouping of the correct object elements in the correct groups, segmenting by noise causes noisy artifacts in

some cases (e.g. Figure 4: *Continuity*, *Gradient Occlusion*). Second, a qualitative inspection reveals how quantitative metrics fail to appropriately judge the goodness of model performance. Since segmentation is done by inference over a noisy set of outputs, and the noise affects the representation of those outputs, the outputs in some datasets (like *Continuity* and *Closure*) are often shifted or misshaped by a small number of pixels, resulting in a disproportionately large drop in quantitative performance, while qualitative performance remains relatively unchanged.

To further understand how elements are grouped in our model, we conducted two additional qualitative analyses: (1) Using UMAP visualization, we inspected model clustering and found that while the AE does not find the correct object binding, the VAE does (Appendix F.1). (2) To understand how well the segmentation method generalizes, we qualitatively evaluated the models on the CelebA dataset (Liu et al., 2015) and found that while the AE generally fails to find a semantically meaningful segmentation mask (and instead opts for more basic image-level features), the VAE often finds a semantically meaningful segmentation of face-hair-background (Appendix B).

### 3.3 NOISE SENSITIVITY

To understand how practically feasible LNS is for a system constrained by compute time or noise magnitude, we analyzed how the amount of noise added in latent space would affect the model's segmentation performance. While enforcing a prior distribution on the model's latent space (as is the case in the VAE) had no benefit in model performance (and perhaps a slight deterioration), it may have had a minor benefit in the across-task best-performing noise consistency. We tested the same models across different levels of noise used for segmentation and found that enforcing the prior $\mathcal{N}(0, \mathbf{I})$ caused the model to consistently perform best at very small levels of noise, while the AE optimal level of noise varied across datasets (Figure 5). This is intuitively not surprising, since the VAE enforces a specific activity magnitude for uninformative units in its prior ($\mathcal{N}(0, \mathbf{I})$), while the AE does not — as such, the AE has no reason to prefer a specific coding magnitude in any given task. To quantify the result, we performed an F-test for the equality of variances, and found that the level of noise that yielded the best results had higher variance for the AE than the VAE ($F = 2994.56, p = 1.11e^{-8}$).

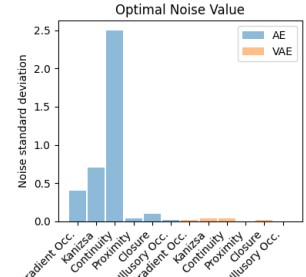

Figure 5: **The optimal noise level for the AE and the VAE across datasets.** The AE performs best with a larger range of noise values, while the VAE consistently prefers low noise values.

The result suggests that enforcing a prior distribution on the representation may have an advantage in stabilizing the amount of noise needed to best segment stimuli across a variety of tasks. That being said, the absolute performance of the AE model did not substantially decrease as a function of noise (Figure 6 left), which means that it is possible that even the AE could yield meaningful segmentation results using a consistently low magnitude of noise.

### 3.4 NUMBER OF TIME STEPS NEEDED

We evaluated model performance as a function of the number of time steps (in other words, the number of samples collected from the model) used for segmentation (Figure 6, **right**). We found that for an appropriate level of noise used in segmentation, both the AE and the VAE models' performance asymptotes quickly, with as few as a handful of time steps. Indeed, reducing the number of time steps used from 80 (Figure 6 **left**) down to as few as 5 barely reduces segmentation accuracy (Figure 6 **middle**).

## 4 CONCLUSION AND DISCUSSION

We present Latent Noise Segmentation (LNS) that allows us to turn a deep neural network that is trained on a generic image reconstruction task into a segmentation algorithm without training any additional parameters, using only independent neural noise. The intuition behind why independent neural noise works for segmentation is that independent noise affects neurons that are selective to a presented stimulus more than those that are not. This idea is not entirely new — noise has

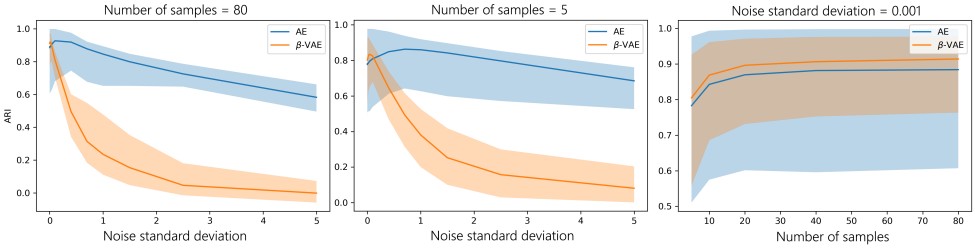

Figure 6: **Sensitivity to Noise Magnitude and Time Steps. Left:** AE and VAE segmentation performance as a function of noise standard deviation, evaluated with 80 time steps. **Middle:** AE and VAE segmentation performance as a function of noise standard deviation, evaluated with 5 time steps. **Right:** AE and VAE segmentation performance as a function of the number of samples for a fixed noise $\sigma = 0.001$. Note the change of $y$-axis scale. Shaded areas show 75th and 25th percentiles over all evaluated segmentation masks.

been shown to theoretically be able to *enhance* the signal of a stimulus when the signal is small (Benzi et al., 1981; Kitajo et al., 2003; Buchin et al., 2016); however, its application to segmentation and perceptual grouping, to the best of our knowledge, is novel.

Importantly, our goal is not to propose a state of the art segmentation algorithm, but rather to understand the role of neural noise and how it can be applied to achieve segmentation and perceptual grouping. We implement LNS and find that it is possible to make pre-trained Autoencoders (AE) and Variational Autoencoders (VAE) segment images. We present the Good Gestalt (GG) datasets — a collection of datasets intended to investigate perceptual organization in humans — and show that LNS is able to explain the formation of 'Good Gestalt' in a neural network model. Our findings give a unified high-level account of the result of many perceptual grouping phenomena in humans: 1) learning encodes an approximation of the training dataset distribution, and 2) neural noise reveals the learned representation in out-of-distribution samples, resulting in certain hallmarks of perception, like effects of illusory contours, closure, continuity, proximity and occlusion. Establishing this link between training data (Appendix E.1), neural noise, and human perception is an important and timely step towards better models of visual representation (Bowers et al., 2022).

Interestingly, model learning rule plays a role in segmentation. We show that a very small amount of independent noise is optimal for segmentation in the case of the VAE, but not for the AE. Similarly, the VAE finds the correct number of objects in the image (Appendix F.1) even when the AE does not. This deviation between the VAE and the AE is interesting, as the VAE has connections with predictive coding (Marino, 2022; Boutin et al., 2020) and the free energy principle (Friston, 2010; Mazzaglia et al., 2022), and has been shown empirical support as a coding principle in primate cortex (Higgins et al., 2021). With both learning rules, segmentation performance asymptotes quickly, suggesting that the time scale of the segmentation in our models is practically feasible for constrained systems.

These results suggest a number of interesting questions for future research. First, in this work our focus has been on revealing a potential mechanism for perceptual grouping and segmentation. While we have demonstrated that the approach partially scales to the case of face segmentation (Appendix B), a relevant question is whether it is possible to scale the approach to improve unsupervised segmentation state-of-the-art (Bear et al., 2023; Engelcke et al., 2022). Second, while we have targeted a large collection of most relevant grouping principles, there are many that we have not tested here. As we have shown for the datasets we have tested, we speculate that most or all phenomena in Gestalt perception can be reproduced as a result of a combination of past experience (learning) and noise in the perceptual system. Of interest is also understanding how LNS interacts with perception under a motion stream, which we have not considered here. Finally, while we do not make any claims about the primate visual system, our results suggest a potential benefit of independent noise in the visual system. Understanding how noise interacts with perception in primate cortex is an active area of research (Destexhe and Rudolph-Lilith, 2012; Miller and Troyer, 2002; Stein et al., 2005; Guo et al., 2018; McDonnell and Ward, 2011), and we hope our results stimulate discussion about the potential benefits of noise for perceptual systems in general.

## REPRODUCIBILITY

Mathematical details about our results are included in Appendix A.1. Full details on our GG datasets are included in Appendix E.1, and full details on model architecture and training are included in Appendix E.2. Code and data will be made available in the camera-ready version of the manuscript.

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

# A APPENDIX

## A.1 OBJECT SEPARABILITY

The Latent Noise Segmentation approach proposed in this work is capable of separating pixels into different objects regardless of their actual values in the input. This fact is a necessary result yielded by a Variational Autoencoder (VAE) pursuing its optimization objective. Here, we formalize the mathematical intuition behind why this is the case.

### A.1.1 BACKGROUND

Rolínek et al. (2019); Zietlow et al. (2021) showed that VAE-based Deep Neural Networks (DNNs) learn to represent local Principal Component (PC) axes. The training process of a VAE aims to minimize the reconstruction loss, part of which can be re-assembled as the stochastic reconstruction loss:

$$\hat{L}_{\text{rec}}\left(\mathbf{x}^i\right) = \mathop{\mathbb{E}}_{\mathbf{z}^i \sim \mathcal{N}(\boldsymbol{\mu}(\mathbf{x}^i), \text{diag}(\boldsymbol{\sigma}^2(\mathbf{x}^i)))} \| \text{Dec}_\theta\left(\boldsymbol{\mu}\left(\mathbf{x}^i\right)\right) - \text{Dec}_\theta\left(\mathbf{z}^i\right) \|^2, \tag{2}$$

where the superscript $i$ denotes the index of the training samples.

We further define a Jacobian matrix to denote the decoded output's derivative with respect to the latent variables:

$$J_i = \left.\frac{\partial \text{Dec}_\theta\left(\mathbf{z}\right)}{\partial \mathbf{z}}\right|_{\mathbf{z}=\boldsymbol{\mu}(\mathbf{x}^i)} \in \mathbb{R}^{D \times d}, \tag{3}$$

where $D = img_x \times img_y \times C$ is the dimension of the images, and $d$ is the number of latent nodes. Rolínek et al. (2019) proved that optimizing equation 2 will promote pairwise orthogonality on the columns of equation 3, which means that the latent variables locally encode orthogonal features in the image space in $\mathbb{R}^D$. Furthermore, Zietlow et al. (2021) analyzed what those pairwise orthogonal directions are. With experiments, they empirically showed that those directions correspond to local PC directions. More concretely, those directions are the PC directions yielded by a Principal Component Analysis (PCA) performed on a subset of all the training samples that lie close to each other in the data space.

Built on the theoretical framework of Rolínek et al. (2019) and Zietlow et al. (2021), we are able to link these results to our context, and provide intuition for why the VAE is able to separate the object and the background.

One relevant technical detail is that the above statement regarding the latent nodes pursuing local local PC directions is derived from a VAE structure with a standardized objective as formulated by Higgins et al. (2017). However, it is easy to check the proof from Rolínek et al. (2019) to see that the same conclusion applies to our case, where the GECO (Rezende and Viola, 2018) loss is used.

Furthermore, although the above-mentioned argument from Rolínek et al. (2019); Zietlow et al. (2021) is for the Jacobians (equation 3) calculated at training samples, we posit that the same conclusion holds for the testing samples. After all, in our setting, the training samples and the testing samples are close to each other. Henceforth, $i$ will be used to denote the index of testing samples.

### A.1.2 MATHEMATICAL INTUITION UNDERPINNING GESTALT PERCEPTION

In the following, we will use $p_1$ to denote a pixel belonging to one object of the image, and $p_2$ to denote a pixel belonging to a different object of the image.

In our segmentation algorithm, we first repeatedly apply noise to the latent units and observe the resultant changes in output pixels. We denote the added noise, at the time step of $n$, to be $\boldsymbol{\xi}_n = (\xi_{n,j})_{j \in \{1,2,\cdots,d\}}^\mathsf{T} \in \mathbb{R}^d$, where the second subscript $j$ denotes the indices of the latent units to which noise is added.

Based on multivariate Taylor expansion, the reconstruction yielded by the latent affected by the noise can be written as:

$$\tilde{\mathbf{x}}_n^i = \text{Dec}\left(\boldsymbol{\mu}\left(\mathbf{x}^i\right) + \boldsymbol{\xi}_n\right) = \text{Dec}\left(\boldsymbol{\mu}\left(\mathbf{x}^i\right)\right) + \sum_{j=1}^{d} \xi_{n,j} \left.\frac{\partial \text{Dec}\left(\mathbf{z}\right)}{\partial z_j}\right|_{\mathbf{z}=\boldsymbol{\mu}\left(\mathbf{x}^i\right)} + o\left(\boldsymbol{\xi}_n\right), \quad (4)$$

in which the superscript $i$ of $\mathbf{x}$ denotes the index of the input image, the subscript $j$ of $z$ denotes the latent index. $\left.\frac{\partial \text{Dec}(\mathbf{z})}{\partial z_j}\right|_{\mathbf{z}=\boldsymbol{\mu}(\mathbf{x}^i)}$ is the $j^{th}$ column of the Jacobian in equation 3. $o\left(\boldsymbol{\xi}_n\right)$ denotes the residual terms of the Taylor expansion.

In our segmentation algorithm, we subtract the reconstruction affected by one set of noise from the reconstruction affected by the previous set of noise. If we use $\boldsymbol{\delta}_n = \left(\delta_{n,j}\right)^{\mathsf{T}}_{j\in\{1,2,\cdots,d\}} \in \mathbb{R}^d$ to represent $\left(\boldsymbol{\xi}_{n+1} - \boldsymbol{\xi}_n\right)$, then we can write the difference between the two consecutive noisy reconstructions as:

$$\Delta\tilde{\mathbf{x}}_n^i = \tilde{\mathbf{x}}_{n+1}^i - \tilde{\mathbf{x}}_n^i = \sum_{j=1}^{d} \delta_{n,j} \left.\frac{\partial \text{Dec}\left(\mathbf{z}\right)}{\partial z_j}\right|_{\mathbf{z}=\boldsymbol{\mu}\left(\mathbf{x}^i\right)} \in \mathbb{R}^D, \quad (5)$$

where we omit the $o(\cdot)$ term, as they are negligible when the perturbing noise is sufficiently small.

Furthermore, we use $\Delta\tilde{\mathbf{x}}_{n,c,p_1}^i \in \mathbb{R}$ to denote the value of the component of $\Delta\tilde{\mathbf{x}}_n^i$ corresponding to the value of the pixel $p_1$ in the channel $c$ and $\Delta\tilde{\mathbf{x}}_{n,c,p_2}^i \in \mathbb{R}$ to denote the value of the component of $\Delta\tilde{\mathbf{x}}_n^i$ corresponding to the value of the pixel $p_2$ in the channel $c$.

Our visual segmentation pipeline separates pixel $p_1$ from pixel $p_2$ by separating the vector $\Delta\tilde{\mathbf{X}}_{p_1}^i = \left(\Delta\tilde{\mathbf{x}}_{n,c,p_1}^i\right)^{\mathsf{T}}_{(n,c)\in\{1,2,\cdots,N-1\}\times\{1,\cdots,C\}} \in \mathbb{R}^{(N-1)\times C}$ from the vector $\Delta\tilde{\mathbf{X}}_{p_2}^i = \left(\Delta\tilde{\mathbf{x}}_{n,c,p_2}^i\right)^{\mathsf{T}}_{(n,c)\in\{1,2,\cdots,N-1\}\times\{1,\cdots,C\}} \in \mathbb{R}^{(N-1)\times C}$.

More concretely, as indicated in Appendix D, the separation between the two vectors are based on the angles they point toward in the $[(N-1)\times C]$-dimensional space. In other words, the case in which the two pixels cannot be separated with our pipeline can only be caused by the case that no matter how we sample our noise, the following condition holds with probability 1.

**Condition for Inseparability.** There exist some constant $r_{p_1,p_2} > 0$ such that, for the given pixel pair $(p_1, p_2)$ and any channel $c$ and any time step $n$: $\Delta\tilde{\mathbf{x}}_{n,c,p_1}^i = r_{p_1,p_2}\Delta\tilde{\mathbf{x}}_{n,c,p_2}^i$.

The equation in the condition for inseparability can be expanded as:

$$\sum_{j=1}^{d} \delta_{n,j} \left.\frac{\partial \text{Dec}\left(\mathbf{z}\right)_{c,p_1}}{\partial z_j}\right|_{\mathbf{z}=\boldsymbol{\mu}\left(\mathbf{x}^i\right)} = r_{p_1,p_2} \sum_{j=1}^{d} \delta_{n,j} \left.\frac{\partial \text{Dec}\left(\mathbf{z}\right)_{c,p_2}}{\partial z_j}\right|_{\mathbf{z}=\boldsymbol{\mu}\left(\mathbf{x}^i\right)} \quad (6)$$

The fact that condition for inseparability holding with probability 1 implies inseparability is intuitive, and we also provide a proof for it in A.1.3.

In the following, we will argue why the condition for inseparability is unlikely to hold with probability 1 given the background in Section A.1.1.

For the condition of inseparability to hold, the Jacobian (in equation 3) and the noise must satisfy one of the following two cases.

The **first case** is when the Jacobian satisfies:

$$\left.\frac{\partial \text{Dec}\left(\mathbf{z}\right)_{c,p_1}}{\partial z_j}\right|_{\mathbf{z}=\boldsymbol{\mu}\left(\mathbf{x}^i\right)} = r_{p_1,p_2} \left.\frac{\partial \text{Dec}\left(\mathbf{z}\right)_{c,p_2}}{\partial z_j}\right|_{\mathbf{z}=\boldsymbol{\mu}\left(\mathbf{x}^i\right)} \quad (7)$$

for some fixed $r_{p_1,p_2}$ for all $j \in \{1,2,\cdots,d\}$ and all channels $c$. Here $\text{Dec}\left(\mathbf{z}\right)_{c,p_2/p_2} \in \mathbb{R}$ is the component of the $D$-dimensional vector $\text{Dec}\left(\mathbf{z}\right)$ that corresponds to the $p_1$ or $p_2$ pixel in channel $c$. In this case, the condition for inseparability trivially holds with probability 1.

The **second case** is where equation 7 does not hold for any fixed $r_{p_1,p_2}$ for some $j \in \{1,2,\cdots,d\}$ and some channel $c$. In this case, the condition for inseparability might still hold.

In the following, we will explain why both cases cannot hold with probability 1 when $p_1$ and $p_2$ are pixels belonging to different objects of the image, which means that they have independent pixel values in the training dataset (see Appendix E.1 for more details).

(1) Why is the first case unlikely to hold?

The argument of the first case boils down to a characterization of the Jacobian in equation 3. A complete theoretical understanding of the Jacobian has yet to be established in the past literature to the best of our knowledge. However, from the conclusions derived from Rolínek et al. (2019) and Zietlow et al. (2021), which we briefly mentioned in Appendix A.1.1, the first case is unlikely to happen as it intuitively goes against the behavior of PCA.

In more detail, as demonstrated in Zietlow et al. (2021), the columns of the Jacobian (equation 3) should point into directions in the $D$-dimensional space that are principal components extracted from a compact cluster of training images in the $D$-dimensional space. In addition, equation 7 implies that all the principal components characterize the pixel values (in each channel) of $p_1$ and $p_2$ to have a strict linear relationship with the same ratio $r_{p_1 p_2}$. However, as introduced in Appendix E.1, the training dataset is generated by assuming the pixel values of $p_1$ and $p_2$ (when they do not belong to the same object) are independent. Thus, it is unlikely to have all local PC directions satisfying equation 7.

For clarity, we provide a diagram in the left panel of Figure 7. Here we project the entire training dataset (denoted by the gray dots) to a 2-dimensional space. Each dimension stands for the pixel value of $p_1$ or $p_2$. The grey dots are arranged in a way to demonstrate the independence of the two pixel values. The orange arrow denotes a potentially learned local PC direction, moving along which the variation of the two pixel values follows a strict linear ratio. Since there are data variances in other directions, it is highly unlikely that all the local PC directions learned from local data points only capture directions of such kind.

On the other hand, we can also develop intuition into why any two pixels belonging to the same object can be inseparable with the right panel of Figure 7. If $p_1$ and $p_2$ are pixels belonging to the same object, the way the dataset is generated (discussed in Appendix E.2) dictates that the training samples projected onto the pixel value space will only stretch in a direction where there exists a strict linear ratio between the values of the two objects. In this case, the learned local PC directions are highly likely to point only in the direction on this 2-dimensional plane, as there exists no data variance along other directions. Such learned local PC directions guarantee that the two pixels of the same object will not be separated in the segmentation process, for it implies the Jacobian satisfies the condition described in equation 7 in the first case.

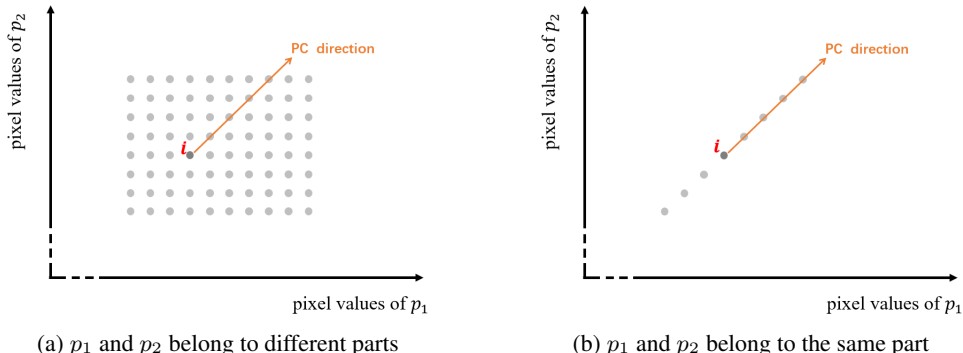

(a) $p_1$ and $p_2$ belong to different parts      (b) $p_1$ and $p_2$ belong to the same part

Figure 7: **(Left)** An example of a direction that all the local PC directions should follow to have $p_1$ and $p_2$ become inseparable. **(Right)** An example of how the local PC directions might look like when they are projected into the pixel value space of two pixels of the same part of the images.

(2) Why can the second case not hold with probability 1?

To understand this case, we first rewrite equation 6 as:

$$\sum_{j=1}^{d} \left[ \left.\frac{\partial \text{Dec}(\mathbf{z})_{c,p_1}}{\partial z_j}\right|_{\mathbf{z}=\boldsymbol{\mu}(\mathbf{x}^i)} - r_{p_1,p_2} \left.\frac{\partial \text{Dec}(\mathbf{z})_{c,p_2}}{\partial z_j}\right|_{\mathbf{z}=\boldsymbol{\mu}(\mathbf{x}^i)} \right] \delta_{n,j} = \sum_{j=1}^{d} C_{c,j}(r_{p_1,p_2})\delta_{n,j} = 0,$$

(8)

in which $C_{c,j}(r_{p_1,p_2})$ is used to summarize the term in the square bracket $[\ \cdot\ ]$. As we are considering the second case, we know that $C_{c,j}(r_{p_1,p_2}) \neq 0$ for at least one $j \in \{1,2,\cdots,d\}$ and one channel $c$. Thus, in the second case, the condition for inseparability can be interpreted as: $\boldsymbol{\delta}_n = (\delta_{n,1}, \delta_{n,2}, \cdots, \delta_{n,d})^\intercal \in \mathbb{R}^d$ lies in a hyperplane as described by equation 8, and the hyperplane has less than $d$ dimensions. Such an event cannot happen with probability 1. This is because $\boldsymbol{\delta}_n$ has a Gaussian distribution in the $d$-dimensional space, implying its probability mass on a hyperplane with lower dimensions is 0.

### A.1.3 WHY PROBABILITY 1 IMPLIES INSEPARABILITY

Here, we give a formal derivation of why the condition for inseparability must hold with probability 1 to make $p_1$ and $p_2$ inseparable. To simplify notation, let us denote the event that equation 6 holds for some time step $n$ some channel $c$ and some positive constant $r_{p_1,p_2}$ to be $\mathcal{A}_{n,c}(r_{p_1,p_2})$. It would require $\mathcal{A}_{n,c}(r_{p_1,p_2})$ to hold for all $n \in \{1,2,\cdots,N-1\}$ and all $c \in \{1,\cdots,C\}$ to make $p_1$ and $p_2$ inseparable, otherwise $\Delta\tilde{\mathbf{X}}_{p_1}^i$ and $\Delta\tilde{\mathbf{X}}_{p_2}^i$, both belonging to $\mathbb{R}^{(N-1)\times C}$, do not point into the same direction, making the two pixels separable.

Intuitively, if the condition for inseparability does not hold with probability 1, then the events $\{\mathcal{A}_n(r_{p_1,p_2})\}_{n\in\{1,2,\cdots,N-1\}}$ are less and less likely to happen simultaneously as $N$ increases, meaning $p_2$ and $p_1$ are always separable given $N$ sufficiently large. Here below, we characterize this intuition rigorously.

Note that $\mathcal{A}_{n,c}(r_{p_1,p_2})$ and $\mathcal{A}_{n+1,c}(r_{p_1,p_2})$ may be dependent on each other, as we can show that $\boldsymbol{\delta}_n$ and $\boldsymbol{\delta}_{n+1}$ are dependent of each other. $\boldsymbol{\delta}_n$ involves the noise pair of $(\boldsymbol{\xi}_n, \boldsymbol{\xi}_{n+1})$ and $\boldsymbol{\delta}_{n+1}$ involves the noise pair of $(\boldsymbol{\xi}_{n+1}, \boldsymbol{\xi}_{n+2})$. Both are affected by $\boldsymbol{\xi}_{n+1}$ and are thus dependent (a more formal explanation of which will be given in the last part of this section). Nonetheless, we have that $\mathcal{A}_{n,c}(r_{p_1,p_2})$ and $\mathcal{A}_{n+2,c}(r_{p_1,p_2})$ are independent of each other, since the former concerns the noise pair of $(\boldsymbol{\xi}_n, \boldsymbol{\xi}_{n+1})$ and the latter concerns the noise pair of $(\boldsymbol{\xi}_{n+2}, \boldsymbol{\xi}_{n+3})$, which are unaffected by each other.

Thus, suppose we arbitrarily choose a channel $c_0$:

$$\mathbb{P}(\text{inseparability}) \leq \mathbb{P}\left(\mathcal{A}_{1,c_0}(r_{p_1,p_2}), \mathcal{A}_{2,c_0}(r_{p_1,p_2}), \cdots, \mathcal{A}_{N-1,c_0}(r_{p_1,p_2})\right) \tag{9a}$$

$$\leq \mathbb{P}\left(\mathcal{A}_{1,c_0}(r_{p_1,p_2}), \mathcal{A}_{3,c_0}(r_{p_1,p_2}), \cdots, \mathcal{A}_{2\lceil\frac{N-1}{2}\rceil-1,c_0}(r_{p_1,p_2})\right) \tag{9b}$$

$$= \prod_{i=1}^{\lceil\frac{N-1}{2}\rceil} \mathbb{P}\left(\mathcal{A}_{2i-1,c}(r_{p_1,p_2})\right) \tag{9c}$$

The independence between $\mathcal{A}_{n,c_0}(r_{p_1,p_2})$ and $\mathcal{A}_{n+2,c_0}(r_{p_1,p_2})$ enables the fully factorized representation in equation 9c. Note that every probability term in equation 9c has the same value, and if they are all smaller than 1 (for all $r_{p_1,p_2} > 0$), the product will decay to zero exponentially (for all $r_{p_1,p_2} > 0$). Thus, we require equation 6 to happen with the probability 1 to achieve inseparability.

The last part of this section demonstrates why $\boldsymbol{\delta}_n$ and $\boldsymbol{\delta}_{n+1}$ are not independent. If $\boldsymbol{\delta}_n$ and $\boldsymbol{\delta}_{n+1}$ are independent, then we must have that the variance of their sum should be equal to the sum of their variances. However,

$$\text{Var}(\boldsymbol{\delta}_{n+1} + \boldsymbol{\delta}_n) = \text{Var}(\boldsymbol{\xi}_{n+2} - \boldsymbol{\xi}_{n+1} + \boldsymbol{\xi}_{n+1} - \boldsymbol{\xi}_n) = \text{Var}(\boldsymbol{\xi}_{n+2} - \boldsymbol{\xi}_n) = 2\boldsymbol{\sigma}_{\boldsymbol{\xi}}^2$$
$$\neq 4\boldsymbol{\sigma}_{\boldsymbol{\xi}}^2 = \text{Var}(\boldsymbol{\xi}_{n+2} - \boldsymbol{\xi}_{n+1}) + \text{Var}(\boldsymbol{\xi}_{n+1} - \boldsymbol{\xi}_n) = \text{Var}(\boldsymbol{\delta}_{n+1}) + \text{Var}(\boldsymbol{\delta}_n),$$

(10)

resulting in a contradiction. With slight abuse of notation, all the terms in equation 10 are $d$-dimensional vectors, and all the operations are component-wise.

### A.1.4 DISCUSSION

With the above, we demonstrated how the underlying local PCA enables VAE to segment the images as we desired. In our experiments, we also show that the Autoencoders can also manage to segment

the images similarly, but that their preferred number of objects is different to the VAEs'. Indeed, qualitatively, the Autoencoder generalizes worse than the VAE (Appendix B). Previous works have drawn a link between PCA and Linear Autoencoders (Plaut, 2018). However, a full theoretical account for the connection between PCA and the more complicated non-linear autoencoder to the best of our knowledge does not yet exist. With the intuitions provided in the experimental observation in this paper, we are at a better place to formulate the connection, which we consider as a meaningful future direction. Furthermore, we have provided a mathematical link between perception in humans, representation in Deep Neural Networks, and semantic segmentation. We hope our work encourages future work in forming rigorous links between these established fields.

### A.1.5 APPLICABILITY TO OTHER GENERATIVE ALGORITHMS

In this work, we have focused primarily on the Variational Autoencoder (VAE) in lieu of other generative models, such as Diffusion models (Sohl-Dickstein et al., 2015; Rombach et al., 2022) or Adversarial Generative (Goodfellow et al., 2014; Creswell et al., 2018) models. There are two primary reasons behind this choice: 1) The VAE formulation allows us to neatly follow the argumentation in Zietlow et al. (2021) and Rolínek et al. (2019) to arrive at a more firm understanding of why Latent Noise Segmentation (LNS) works; 2) the mathematical and computational links between the VAE formulation and other biologically plausible coding and learning schemes such as predictive coding (Boutin et al., 2020; Marino, 2022) and the Free Energy Principle (Friston, 2010; Mazzaglia et al., 2022) are clearer than for other generative algorithms to the best of our knowledge.

That being said, the fact that the Autoencoder works relatively well for segmentation using LNS is interesting, and raises the question of whether other model architectures could also perform well. We suggest that future work could empirically evaluate LNS on other model classes, such as Latent Diffusion models (Rombach et al., 2022) to understand whether the mathematical intuition behind why LNS works also generalizes beyond the simple Autoencoder model class to any model class that learns meaningful representations about images.

# B    GENERALIZATION TO MORE NATURALISTIC STIMULI

To test the generalization and potential scalability of our models, we evaluated them qualitatively on the CelebA dataset (Liu et al., 2015). The dataset consists of a large variety of faces in different contexts, lighting conditions, and backgrounds. We trained the model on the CelebA training dataset as provided, and tested on the first 100 samples of the testing dataset.

We found that the VAE model successfully finds a desired face-hair-background segmentation mask relatively often, when prompted on 3 output clusters (Figure 8). In some cases the clustering is coherent but not semantically aligned with the desired output — we suspect that in the CelebA dataset, lighting condition forms one of the strongest local PC components, and that this leads to a semantically less meaningful cluster on occasion. This effect is seen more strongly in the case of the AE, which appears to primarily follow a lighting-related clustering (Figure 9).

Model: VAE, dataset: CelebA

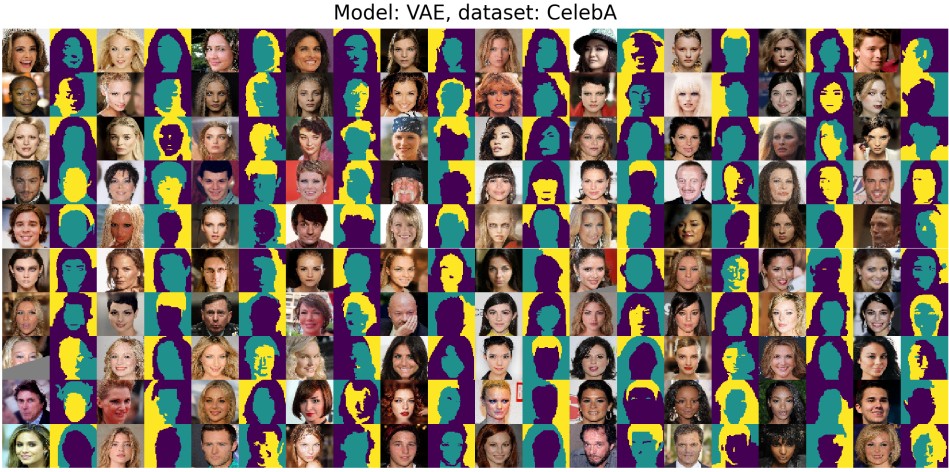

Figure 8: **All tested Variational Autoencoder outputs for a single seed of the CelebA dataset.** Inputs and outputs are displayed as horizontal pairs, with the input being the left image, and the model output being the right image.

Model: AE, dataset: CelebA

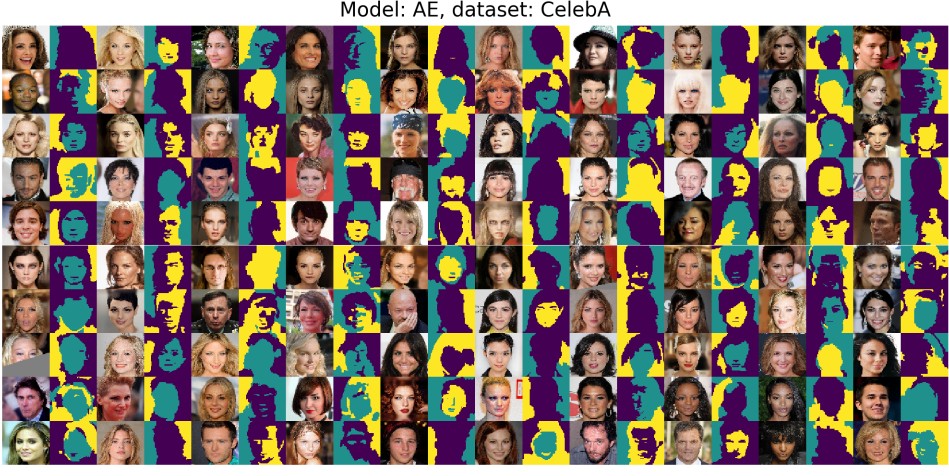

Figure 9: **All tested Autoencoder outputs for a single seed of the CelebA dataset.** Inputs and outputs are displayed as horizontal pairs, with the input being the left image, and the model output being the right image.

## C  CONTROL EXPERIMENTS

### C.1  ADDITIONAL CONTROL EXPERIMENT RESULTS

We tested two high-performing unsupervised object segmentation models from the literature: Genesis (Engelcke et al., 2020a) and Genesis-v2 (Engelcke et al., 2022). We show quantitative results in Table 2. While Genesis performs quantitatively relatively well, a closer inspection of the actual segmentation performance of the model shows that with the exception of the Closure and Continuity datasets, it fails to find the correct Gestalt grouping of the stimuli 10, often opting to follow the color value cues in the testing dataset. The Genesis-v2 performs generally worse than Genesis, failing on all datasets both quantitatively and qualitatively with perhaps the exception of the Continuity dataset.

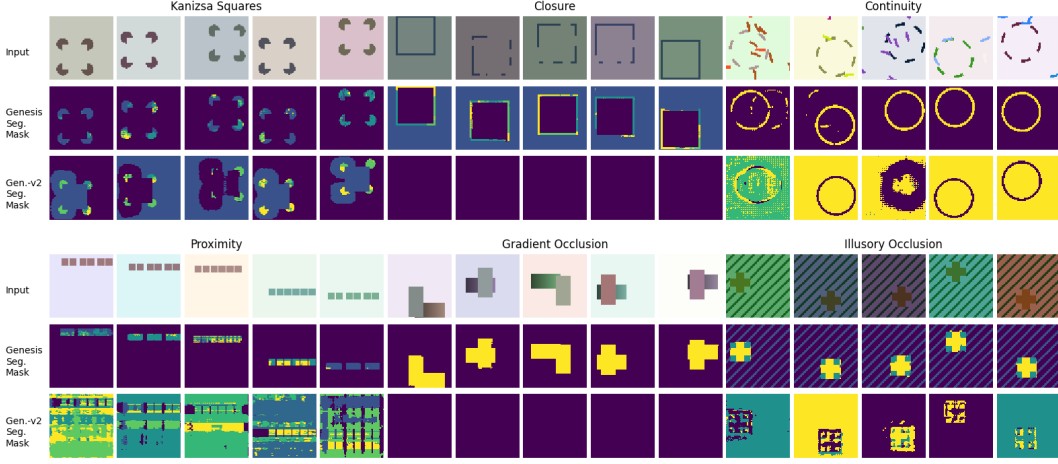

Figure 10: **Control Model Segmentation Mask Examples.** The first row shows inputs, the second row shows Genesis segmentation masks, and the third row shows Genesis-v2 segmentation masks. Randomly selected examples from the **Kanizsa Squares**, **Closure**, **Continuity**, **Proximity**, **Gradient Occlusion**, and **Illusory Occlusion** datasets. The specific color assignment to different object identities is arbitrary (for example, whether the model assigns the identity represented by yellow as the background, or purple, is not a meaningful distinction).

### C.2  IMPLEMENTATION DETAILS AND CONVERGENCE

To ensure that the baseline failure is not due to an implementation error or a failure in convergence, we take two precautions: 1) We follow the official implementation of Genesis and Genesis-v2 (Engelcke et al., 2022; 2020a;b) (GPL v3 license); 2) We verify convergence by visualizing model reconstructions on a separate validation set with samples that were not shown during training 11. Furthermore, we follow the hyperparameter choices reported in the original papers ($5e^5$ training iterations with batch size $64$, $K = 5$). Models were trained using an RTX 4090 GPU. Genesis-v2 trained for approximately 30 GPU-hours per dataset, while Genesis trained for approximately 20 GPU-hours per dataset, for a combined total of approximately 300 GPU-hours.

### C.3  NOISY AE/VAE CONTROL EXPERIMENT IMPLEMENTATION DETAILS

The noise standard deviation applied to the reconstructions is $0.3235$, the mean of the best noise values for the AE and VAE. For clustering we used Agglomerative Clustering with the Euclidean metric and Ward linkage.

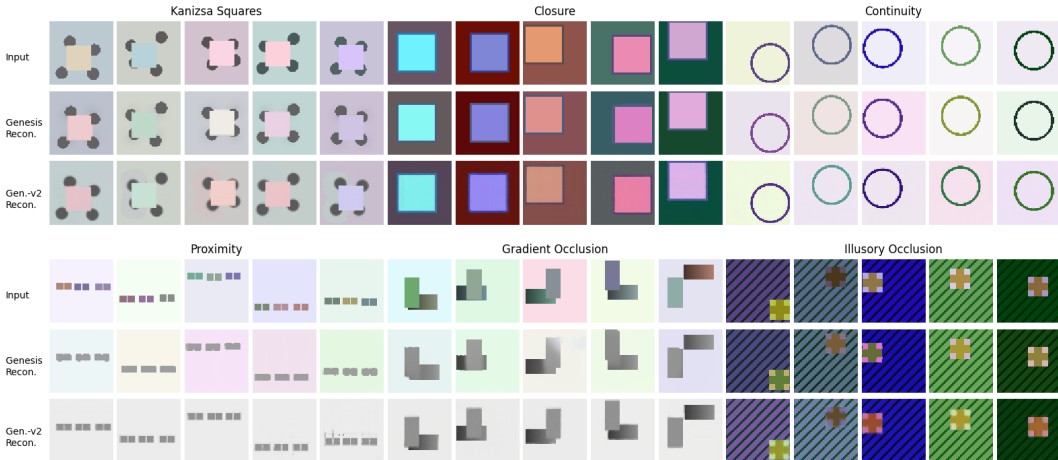

Figure 11: **Control Model Reconstruction Examples.** The first row shows inputs, the second row shows Genesis reconstructions, and the third row shows Genesis-v2 reconstructions. Randomly selected examples from the **Kanizsa Squares**, **Closure**, **Continuity**, **Proximity**, **Gradient Occlusion**, and **Illusory Occlusion** validation datasets. The validation samples come from the same distribution as the training data, but the specific samples are never seen in training.

## D    CLUSTERING

We used Agglomerative Clustering (Pedregosa et al., 2011) with the Euclidean metric and Ward linkage. We selected the desired number of clusters on an image-by-image basis. The desired number of clusters is the number of co-varying parts in the visual scene. For example, for an image in the GG Proximity dataset, if it contains one group of 6 squares, then the desired number of clusters is 2, corresponding to the co-varying squares and the background; if it contains 3 groups of 2 squares, then the desired number of clusters is 4, corresponding to the three groups of co-varying squares and the background.

Prior to applying the agglomerative clustering algorithm, we normalize $\Delta\tilde{\mathbf{X}}$ along its last dimension, so the vectors (with a size of $[\mathrm{C} \times (N-1)]$) have the same norm.

### D.1    COMPARISON OF DIFFERENT CLUSTERING ALGORITHMS

We conducted a control experiment to test whether the specific clustering we used is a critical part of why LNS works. We tested three clustering algorithms: Agglomerative Clustering ((Pedregosa et al., 2011) with the Euclidean metric and Ward linkage; Agglomerative Clustering with the Euclidean metric and Complete linkage; and K-means clustering. We find that while the choice of clustering algorithm can affect results (Ward linkage outperforms Complete linkage), the specific algorithm is not a critical part of LNS (K-means achieves similar or better performance than Agglomerative Clustering with the Ward linkage). This result supports the conclusion that the key intuition behind LNS is that it allows the model to extract a meaningful clusterable representation, that can then be clustered using various clustering algorithms.

| Model | Kanizsa | Closure | Continuity | Proximity | Gradient Occ. | Illusory Occ. |
|---|---|---|---|---|---|---|
| AE+Agg | $0.859_{\pm 0.008}$ | $0.766_{\pm 0.008}$ | $0.552_{\pm 0.008}$ | $\mathbf{0.996}_{\pm 0.001}$ | $\mathbf{0.926}_{\pm 0.009}$ | $0.994_{\pm 0.002}$ |
| VAE+Agg | $\mathbf{0.871}_{\pm 0.005}$ | $0.795_{\pm 0.009}$ | $\mathbf{0.593}_{\pm 0.012}$ | $0.943_{\pm 0.010}$ | $0.918_{\pm 0.002}$ | $0.974_{\pm 0.003}$ |
| AE+KMeans | $0.854_{\pm 0.003}$ | $0.803_{\pm 0.001}$ | $0.553_{\pm 0.004}$ | $0.995_{\pm 0.000}$ | $0.922_{\pm 0.006}$ | $\mathbf{0.998}_{\pm 0.000}$ |
| VAE+KMeans | $0.866_{\pm 0.003}$ | $\mathbf{0.852}_{\pm 0.006}$ | $0.589_{\pm 0.006}$ | $0.921_{\pm 0.005}$ | $0.924_{\pm 0.003}$ | $0.986_{\pm 0.001}$ |
| AE+Agg' | $0.557_{\pm 0.007}$ | $0.296_{\pm 0.006}$ | $0.322_{\pm 0.004}$ | $0.929_{\pm 0.004}$ | $0.702_{\pm 0.004}$ | $0.520_{\pm 0.027}$ |
| VAE+Agg' | $0.457_{\pm 0.008}$ | $0.324_{\pm 0.004}$ | $0.302_{\pm 0.004}$ | $0.752_{\pm 0.012}$ | $0.594_{\pm 0.009}$ | $0.438_{\pm 0.006}$ |

Table 2: Model results for different clustering algorithms (ARI $\pm$ standard error of the mean). Agg = the agglomerative clustering algorithm from `sklearn` with `metric` being "euclidean" and `linkage` being "ward". Agg' = the agglomerative clustering algorithm from `sklearn` with `metric` being "euclidean" and `linkage` being "complete". KMeans = the K-means algorithm from `sklearn`. These scores are obtained for the optimal noise level and number of samples found for Agg.

# E  LNS IMPLEMENTATION DETAILS

## E.1  GG DATASETS

Here we introduce the design of the GG datasets. Each image of the datasets contains several parts including one or several (groups of) objects and a background. In the training dataset, the pixel values belonging to the same part co-vary with each other in the same channel while the pixel values belonging to different parts take independent values in the same channels. Such a scheme is expected to abstract, at a high level, the statistical structure of realistic visual scenes, which is based on basic optics:

$$L = I \times R, \tag{11}$$

where $L$ is luminance, $I$ is illuminance, and $R$ is reflectance. The rationale behind the above-mentioned relationship of different pixels is explained below.

The images in the datasets all have 3 channels. For simplicity, let us only consider one of the channels. The pixel values of one channel are the luminances of the reflected light that has the same wavelength. If two pixels belong to the same object, the ratio between their reflectances is fixed. This is because, in real life, the reflectance ratio between the different parts of the same object is fixed, as it is determined by the material. On the other hand, the reflectances of two pixels belonging to two different objects that can appear in a visual scene are independent. Furthermore, we assume that all the pixels of one object are exposed to the same lighting condition, namely the same illuminance, which is a reasonable simplifying assumption for simple visual scenes in real life. With the above, one can draw the conclusion that the luminances (pixel values) of different pixels belonging to the same object should keep a fixed ratio between each other throughout the training dataset, the luminances (pixel values) of different pixels belonging to different objects should be independent. Moreover, the pixel values of the same pixel in different channels are independent. Because the object might be in different lighting conditions in different visual scenes, and the illuminances of different channels under different lighting conditions should be independent.

Here, we describe how the different GG datasets are constructed. We encourage the reader to revisit Figures 2 and 3 while reading the following for a better understanding.

**Kanizsa Squares** contains three parts: a square, four circles, and a background. The four circles always have the same pixel values. Their relative positions are fixed in the testing dataset but not in the training dataset. In the testing dataset, although the square and the background have the same color, the model is expected to separate these two parts since it is primed by the training samples to do so. Such expectation is reflected in the segmentation masks (Figure 3).

**Closure** contains two parts, one outlined square and a background. In the testing dataset, the interior of the outlined square and the background share the same pixel values, and the outlines of the squares are sometimes occluded partially. The pixel value ratio of the outline and the square interior is kept the same as in the training dataset. The model is expected to perceive the squares together with the outlines.

**Continuity** contains two parts in the training dataset, a circle and a background. In the testing dataset, the circle is fragmented, and several randomly generated fragments are placed at arbitrary positions. The model should perceive the existence of a circle in the testing dataset despite these disturbances.

**Proximity** contains either 2 parts or 4 parts. In the former case, it contains a background one group of 6 squares with an interspace of 1 pixels. In the latter case, it contains a background and 3 groups of 2 squares. The interspaces between different squares within the same group are 1 pixel wide, and the interspaces between different groups are at least 3 pixel wide so that the two cases cannot be confounded. In the testing dataset, all the squares have the same color. However, the model is expected to group different squares together correctly given the interspaces.

**Gradient Occlusion** contains three parts, one tall rectangle with homogeneous color, one wide rectangle with gradient color, and a background. In the testing dataset, the homogeneously-colored rectangle always occludes the gradient-colored rectangle. However, the model is expected to group together the pixels belonging to the gradient-color rectangle, even if they do not have the same pixel values and are sometimes separated by the homogeneously-colored rectangle.

**Illusory Occlusion** contains 2 parts, one striped background, and one striped square. In the testing dataset, part of the background and the foreground object have the same pixel value. However, the model is expected to identify the foreground-background structure.

For code to generate the GG datasets, please refer to `create_datasets.py` in [our git repo will be here in the camera-ready version of the manuscript]. The training set in our experiments contained $30,000$ images, the validation set contained $300$ images that were not in the training set, and the test set contained $100$ images.

## E.2  DEEP NEURAL NETWORKS

| Hyperparameter | Value |
|---|---|
| Optimizer | Adam |
| Learning rate | $5e^{-5}$ |
| Adam betas | (0.9, 0.999) |
| Batch size | 64 |
| Model latent dim | 15(GG)/50(CelebA) |
| Training iterations | $1.1e^6$ |
| AE loss function | MSE |
| VAE loss function | GECO* (Rezende and Viola, 2018)] |
| GECO g_goal | 0.0006(GG)/0.0125 (CelebA) |

Table 3: Autoencoder and VAE hyperparameters used in the experiments. * Adapted from the implementation by Engelcke et al. (2022).

| | Layer | Output Dimension | Kernel size | Stride | Padding | Activation function |
|---|---|---|---|---|---|---|
| | Conv2d | 32 | 3 | 1 | 1 | ReLU |
| | Conv2d | 64 | 4 | 2 | 1 | ReLU |
| | Conv2d | 64 | 4 | 2 | 1 | ReLU |
| Enc | Conv2d | 128 | 4 | 2 | 1 | ReLU |
| | Conv2d | 128 | 4 | 2 | 1 | ReLU |
| | Conv2d | 256 | 4 | 1 | 0 | ReLU |
| | Linear | 128 | - | - | - | ReLU |
| | Linear | 15 | - | - | - | - |
| Dec | Inverse of the encoder | | | | | |

Table 4: Autoencoder and VAE architecture.

# F  CLUSTERABILITY OF REPRESENTATIONS

## F.1  AES, BUT NOT VAES, PERCEIVE THE WRONG NUMBER OF CLUSTERS

In our Segmentation algorithm, the expected number of clusters is provided on an image-by-image basis as *a priori* information for the model. In most cases, the expected number of clusters coincides with the actual clusters spontaneously formed by $\Delta\tilde{\mathbf{X}}_i$, which can be qualitatively verified with a UMAP visualization (McInnes et al., 2018). However, there are exceptions where $\Delta\tilde{\mathbf{X}}_i$ is not readily grouped into the expected numbers of clusters. We observe such exceptions in segmentation using AE on the GG Proximity dataset (when there only exists 1 group of 6 squares) and the GG Illusory Occlusion dataset. Here, we present the UMAP visualization for them using one example per dataset. Note that while not shown here, we observe that the qualitative pattern of the AE being unable to capture the expected number of clusters generally holds for these two cases.

In Figures 12b and 12c, we show the distribution of the vectors corresponding to each pixel in the $[C \times (N-1)]$-dimensional space where clustering is performed, when the models are confronted with an image from the GG Proximity dataset with 1 group of 6 squares. The image and the expected target mask are shown in Figure 12a. In this case, the AE tends to separate the pixels belonging to the squares into 3 clusters, perhaps as a result of being affected by training samples where there exist 3 groups of 2 squares; while the VAE tends to perceive the right number of clusters. However, when given the expected number of clusters, which is 2 in this case, the AE managed to nevertheless yield the correct segmentation mask (indicated by the inset in Figure 12b).

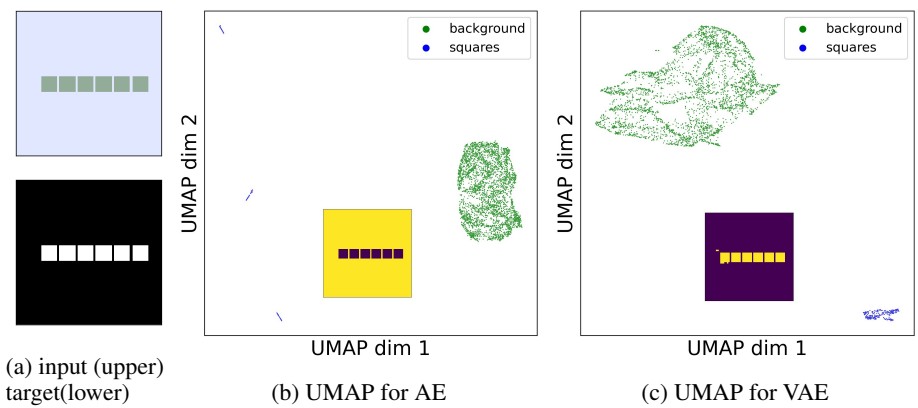

(a) input (upper)
target(lower)

(b) UMAP for AE

(c) UMAP for VAE

Figure 12: The AE perceives the wrong numbers of clusters for the GG Proximity dataset with 1 group of 6 squares. (a) contains the input image and the expected mask. (b) and (c) show the visualization of $\Delta\tilde{\mathbf{X}}^i$. Each dot corresponds to a pixel in the image. The visualization is made by using UMAP to reduce the $[C \times (N-1)]$-dimensional vector corresponding to each pixel to 2 dimensions. The color of the dots indicates which part the pixels correspond to. The insets are the corresponding segmentation masks generated by the AE/VAE model.

A similar situation happens also for the GG Illusory Occlusion dataset. As shown in Figure 13, the AE tends to separate the background from the background stripes. We would expect these two parts to be grouped together, since they always co-vary with each other. Nonetheless, given the expected number of clusters, the AE manages to ultimately segment correctly.

## F.2  LATENT NOISE SEGMENTATION FORMS CLUSTERABLE REPRESENTATIONS

We further show the UMAP visualization of $\Delta\tilde{\mathbf{X}}_i$ to demonstrate that LNS forms a meaningful clusterable representation of the pixels in general. $\Delta\tilde{\mathbf{X}}_i$ is first reduced to $imag_x \times imag_y \times 2$ with UMAP performed on its last two dimensions, which contains $[C \times (N-1)]$-dimensional representations for each pixels. The resulting UMAP representation, accordingly, contains 2-dimensional representations for all the pixels, which we visualize in Figure 15. The position of each dot shows the 2-dimensional representation for each pixel, and we differentiate between different parts of the

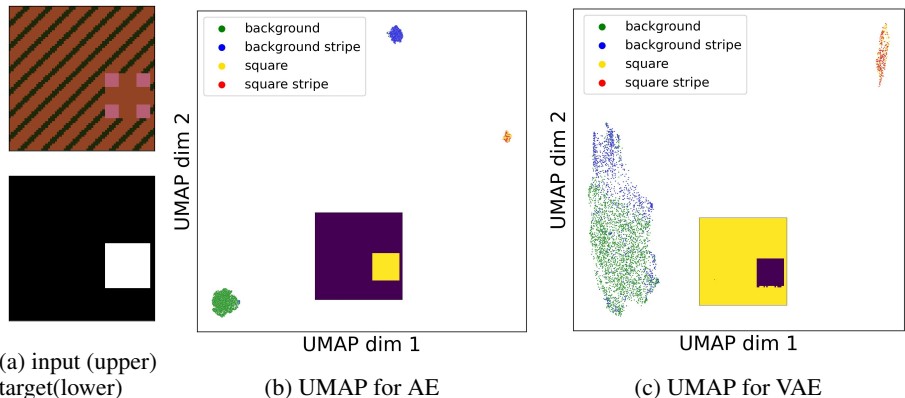

(a) input (upper)
target(lower)

(b) UMAP for AE

(c) UMAP for VAE

Figure 13: The AE perceives the wrong number of clusters for the GG Illusory Occlusion Dataset. Please refer to the caption of Figure 12.

images with different colors. The UMAP results in in Figure 15 are generated while segmenting the samples in Figure 4.

### F.3 MODEL TEST RECONSTRUCTIONS

To support our control experiment, we also visualize the model output reconstructions (Figure 14). In accordance with the control experiments, we find that the outputs in some cases contain some information about the identity of the objects, even when it does not veridically exist in the input image. However, as reported in Table 2, the amount of information in the reconstructions is not as high as it is when using Latent Noise Segmentation. This is because of the tension between reconstructing an image well (which would lead to poor segmentation without LNS in test samples), and segmenting the image components based on their color values. In other words, the fact that the model is still able to segment in some cases can be understood as an artefact caused by the small size of our model and imperfect reconstruction performance. The training process implicitly optimizes toward the success of LNS segmentation, but against the success of direct reconstruction segmentation.

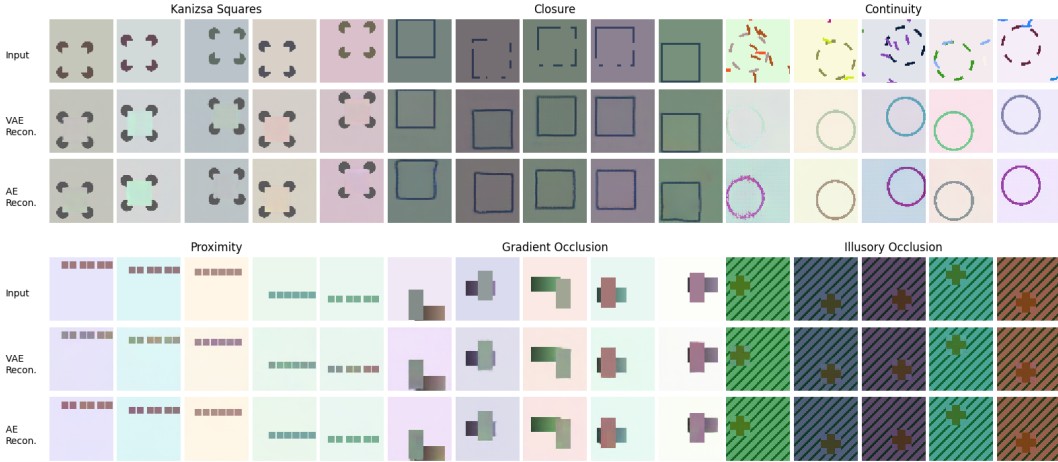

Figure 14: **AE and VAE Model Reconstruction Examples.** The first row shows inputs, the second row shows VAE reconstructions, and the third row shows AE reconstructions. Randomly selected examples from the **Kanizsa Squares**, **Closure**, **Continuity**, **Proximity**, **Gradient Occlusion**, and **Illusory Occlusion** test datasets.

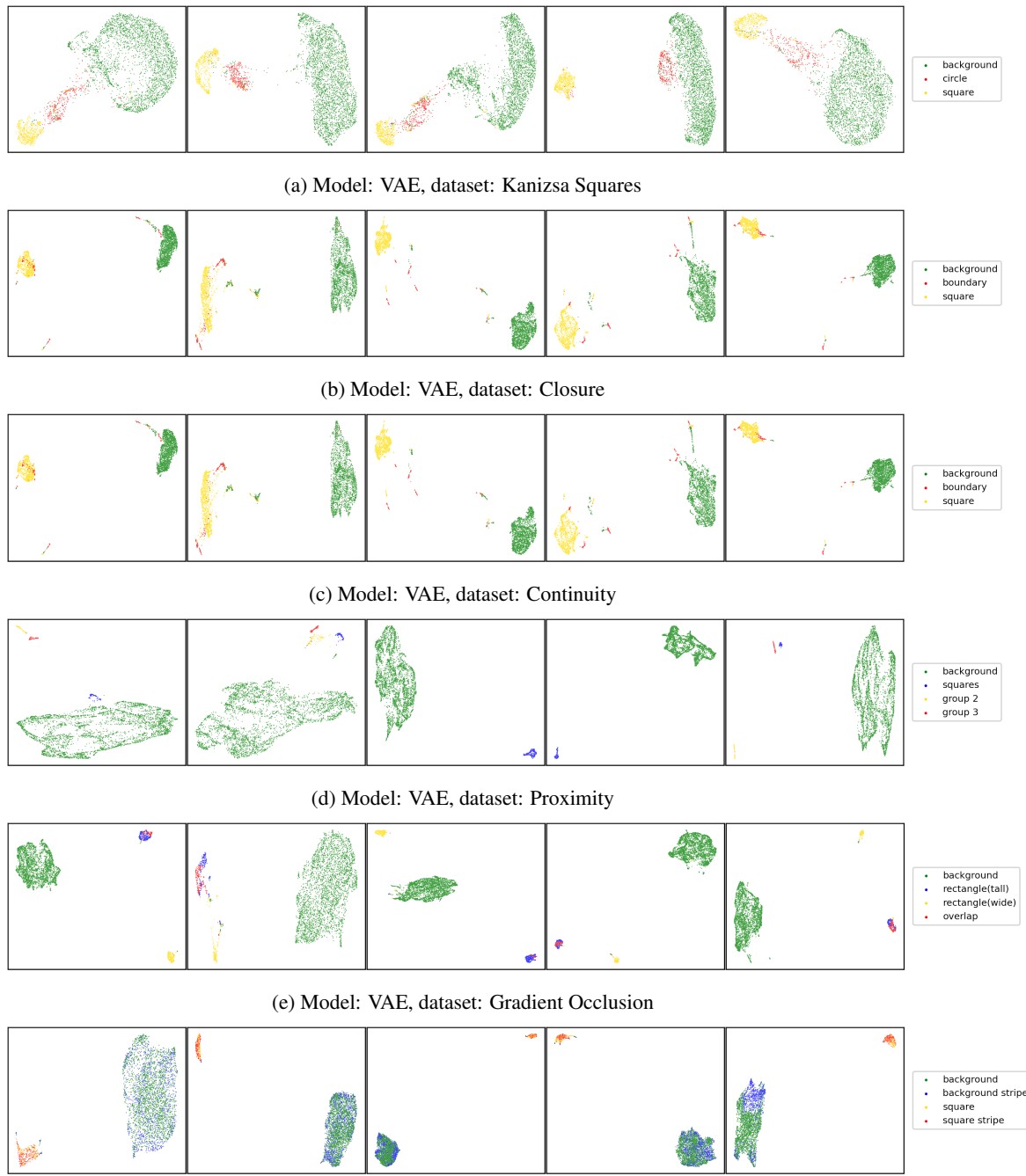

(a) Model: VAE, dataset: Kanizsa Squares

(b) Model: VAE, dataset: Closure

(c) Model: VAE, dataset: Continuity

(d) Model: VAE, dataset: Proximity

(e) Model: VAE, dataset: Gradient Occlusion

(f) Model: VAE, dataset: Illusory Occlusion

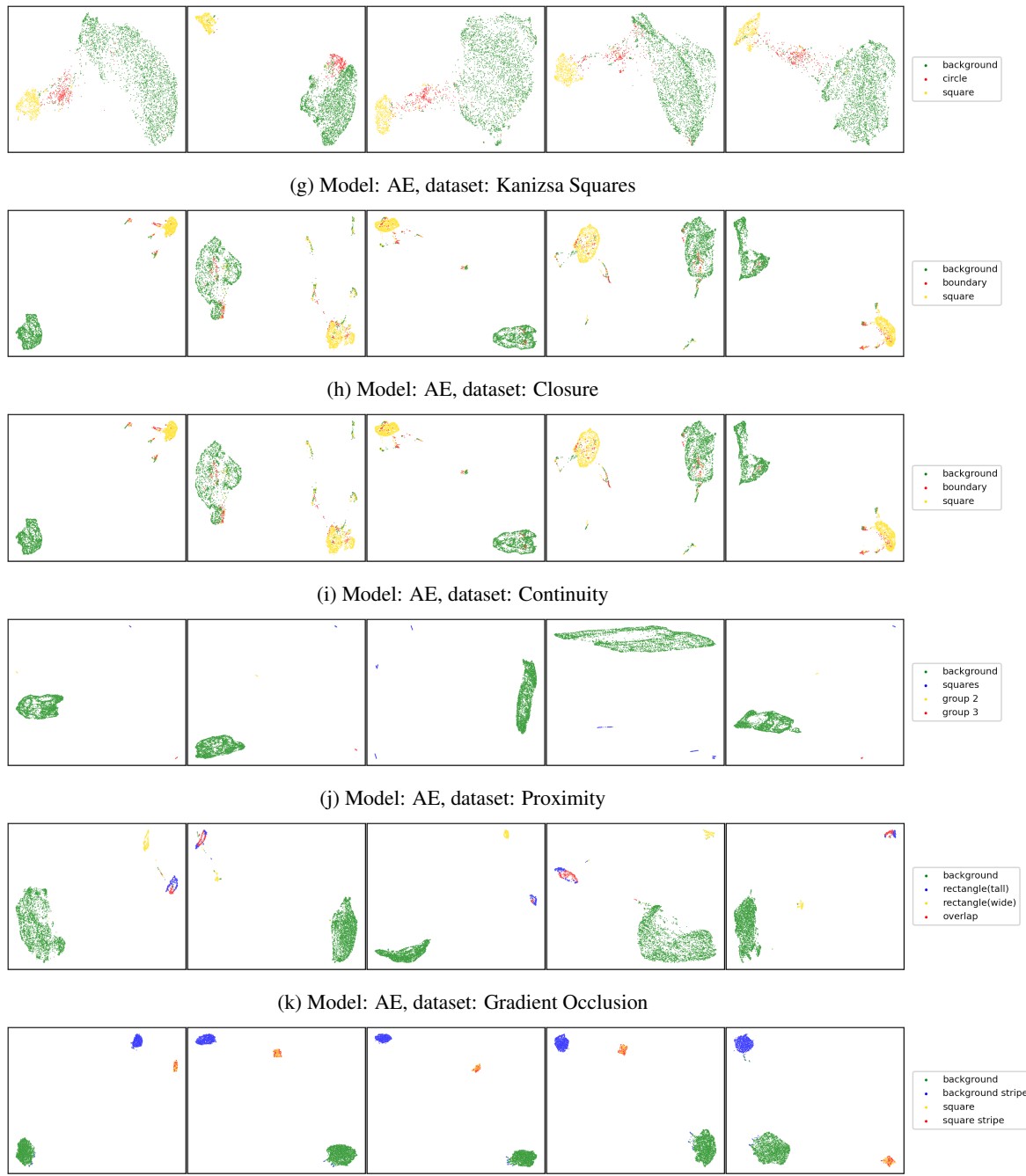

(g) Model: AE, dataset: Kanizsa Squares

(h) Model: AE, dataset: Closure

(i) Model: AE, dataset: Continuity

(j) Model: AE, dataset: Proximity

(k) Model: AE, dataset: Gradient Occlusion

(l) Model: AE, dataset: Illusory Occlusion

Figure 15: Visualization of $\Delta\tilde{\mathbf{X}}^i$ with UMAP

## F.4 ADDITIONAL MODEL OUTPUTS

Here, we plot all model outputs for our testing set for a single seed for both model architectures (AE and VAE) to qualitatively confirm that our models capture the meaningful and relevant desired properties in our datasets, and that our qualitative model performance holds in general, and not only in a few hand-picked examples.

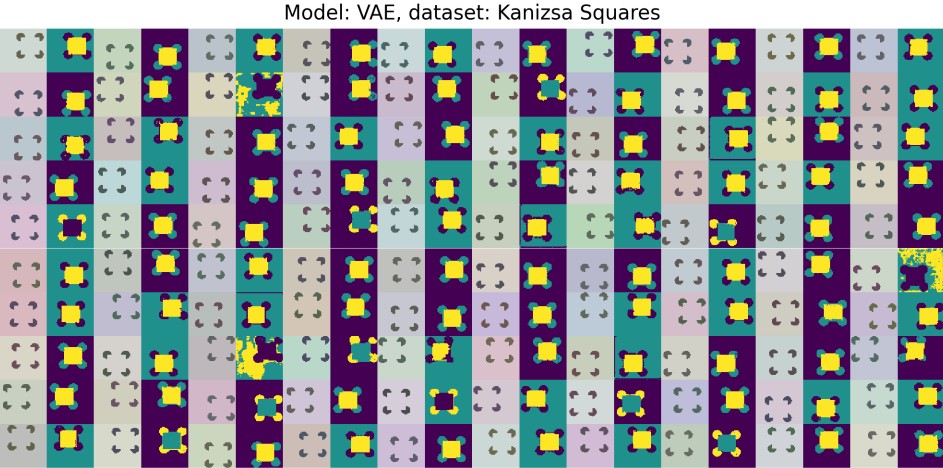

Figure 16: **All Variational Autoencoder outputs for a single seed of the GG Kanizsa Squares dataset.** Inputs and outputs are displayed as horizontal pairs, with the input being the left image, and the model output being the right image.

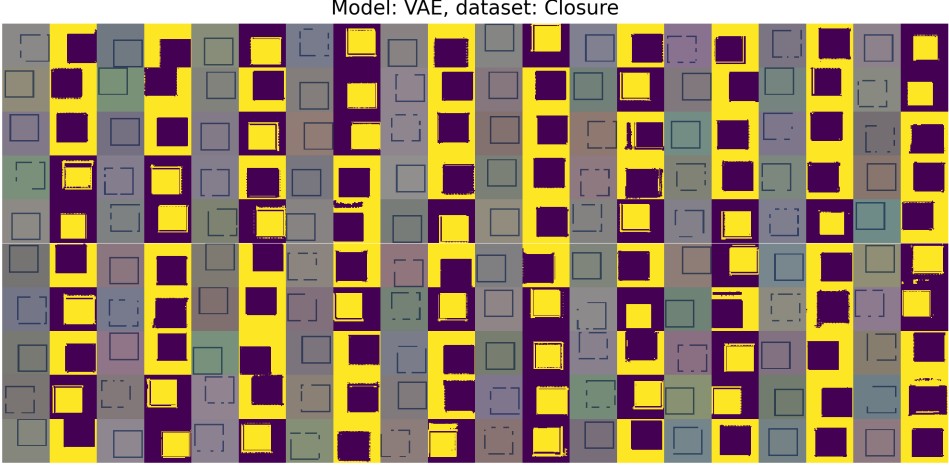

Figure 17: **All Variational Autoencoder outputs for a single seed of the GG Closure dataset.** Inputs and outputs are displayed as horizontal pairs, with the input being the left image, and the model output being the right image.

Model: VAE, dataset: Continuity

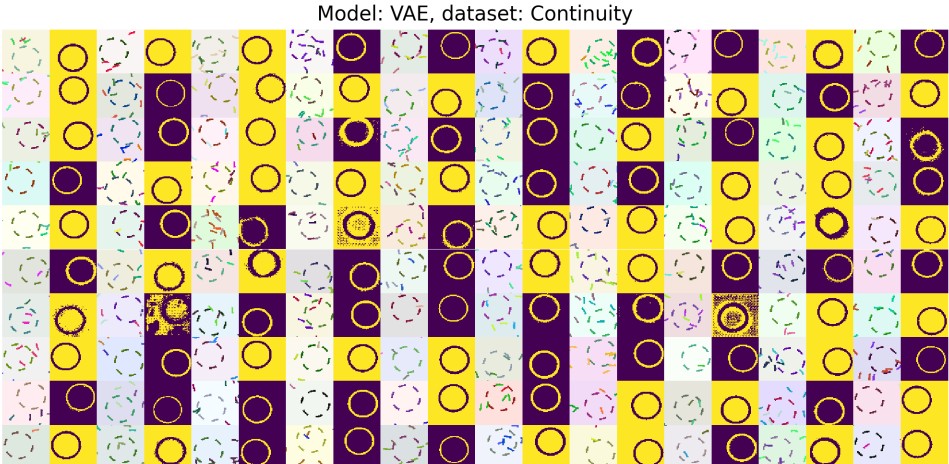

Figure 18: **All Variational Autoencoder outputs for a single seed of the GG Continuity dataset.** Inputs and outputs are displayed as horizontal pairs, with the input being the left image, and the model output being the right image.

Model: VAE, dataset: Proximity

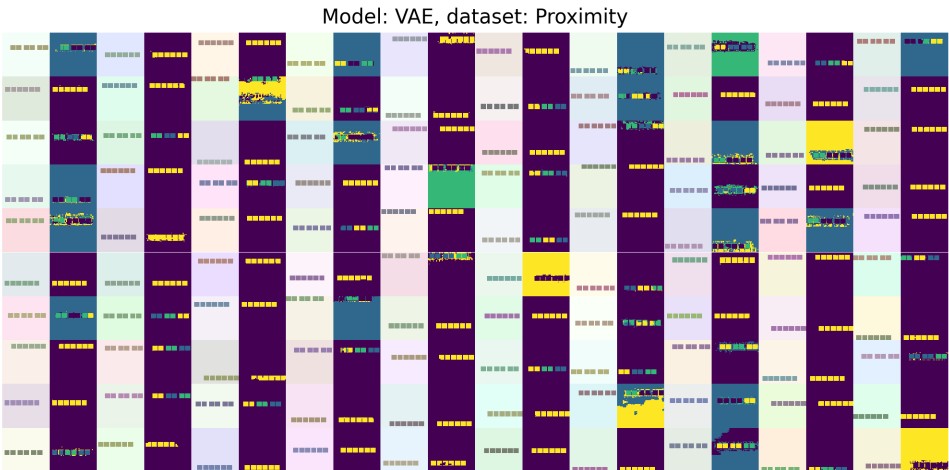

Figure 19: **All Variational Autoencoder outputs for a single seed of the GG Proximity dataset.** Inputs and outputs are displayed as horizontal pairs, with the input being the left image, and the model output being the right image.

Model: VAE, dataset: Gradient Occlusion

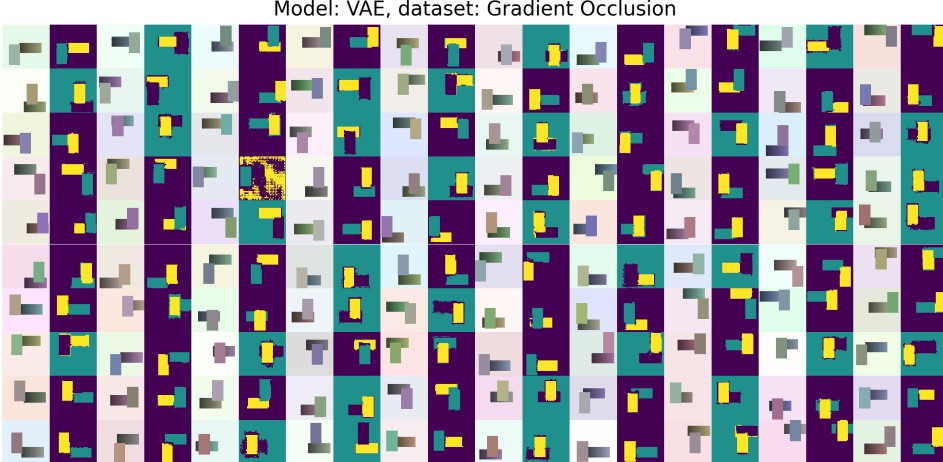

Figure 20: **All Variational Autoencoder outputs for a single seed of the GG Gradient Occlusion dataset.** Inputs and outputs are displayed as horizontal pairs, with the input being the left image, and the model output being the right image.

Model: VAE, dataset: Illusory Occlusion

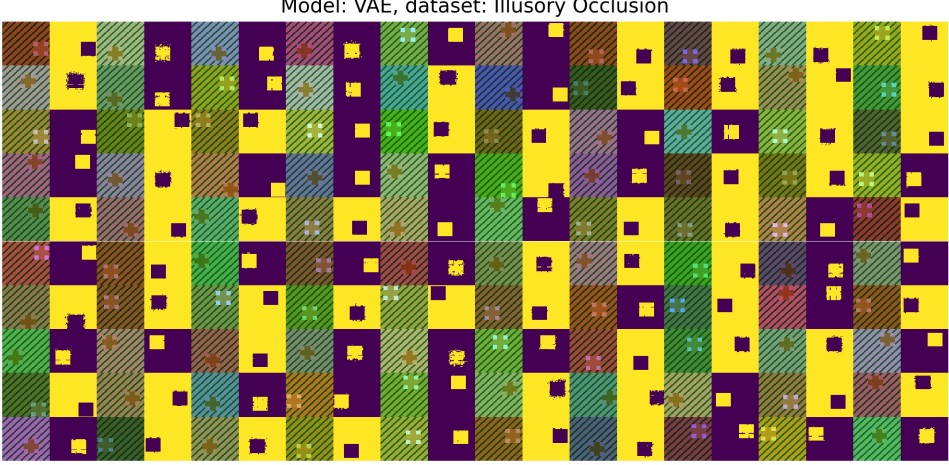

Figure 21: **All Variational Autoencoder outputs for a single seed of the GG Illusory Occlusion dataset.** Inputs and outputs are displayed as horizontal pairs, with the input being the left image, and the model output being the right image.

Model: AE, dataset: Kanizsa Squares

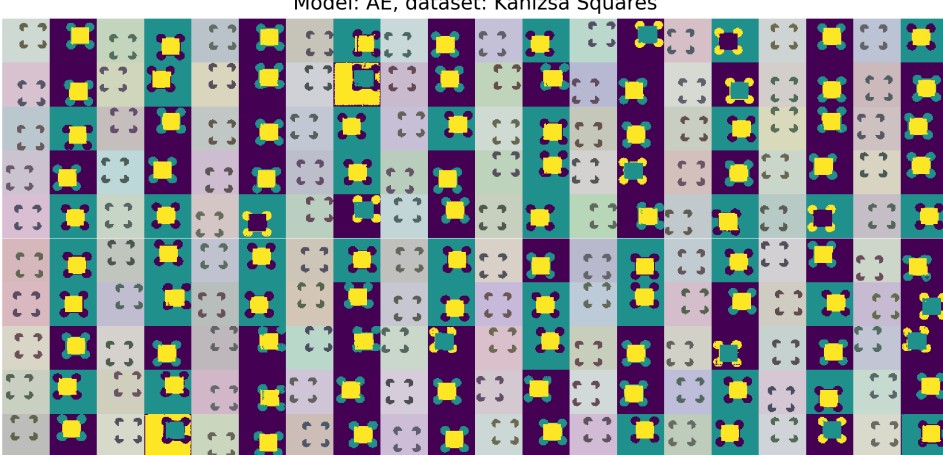

Figure 22: **All Autoencoder outputs for a single seed of the GG Kanizsa Squares dataset.** Inputs and outputs are displayed as horizontal pairs, with the input being the left image, and the model output being the right image.

Model: AE, dataset: Closure

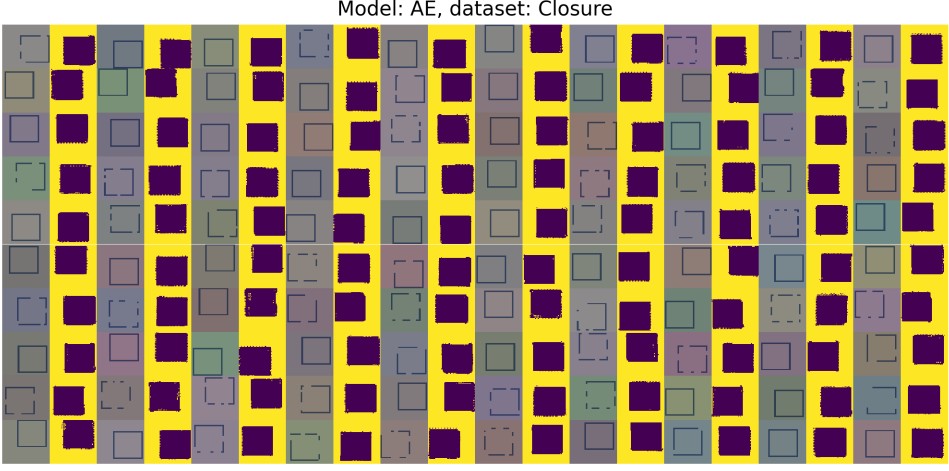

Figure 23: **All Autoencoder outputs for a single seed of the GG Closure dataset.** Inputs and outputs are displayed as horizontal pairs, with the input being the left image, and the model output being the right image.

Model: AE, dataset: Continuity

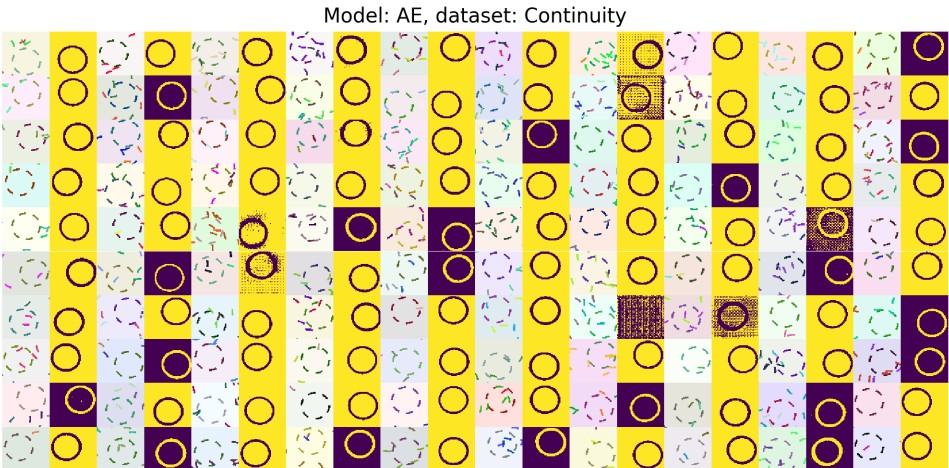

Figure 24: **All Autoencoder outputs for a single seed of the GG Continuity dataset.** Inputs and outputs are displayed as horizontal pairs, with the input being the left image, and the model output being the right image.

Model: AE, dataset: Proximity

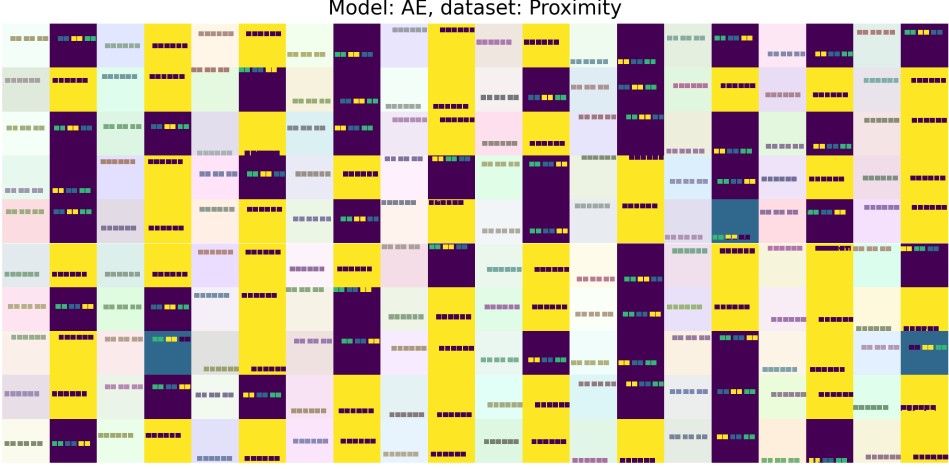

Figure 25: **All Autoencoder outputs for a single seed of the GG Proximity dataset.** Inputs and outputs are displayed as horizontal pairs, with the input being the left image, and the model output being the right image.

Model: AE, dataset: Gradient Occlusion

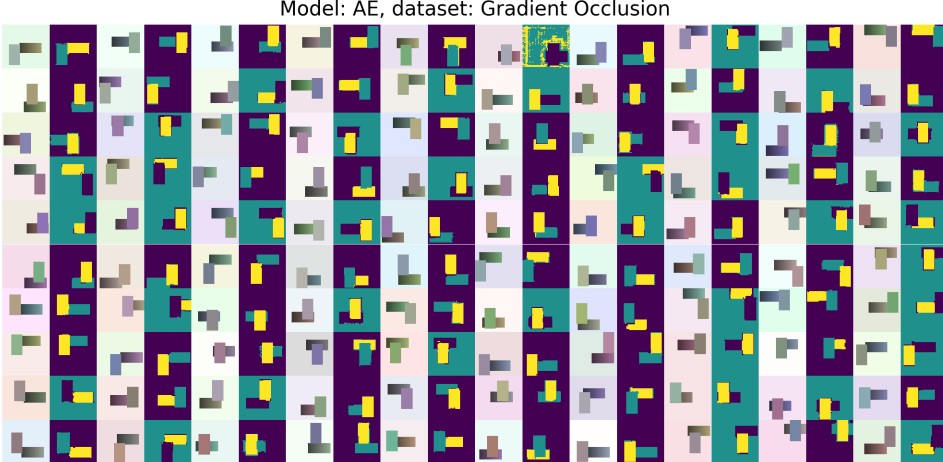

Figure 26: **All Autoencoder outputs for a single seed of the GG Gradient Occlusion dataset.** Inputs and outputs are displayed as horizontal pairs, with the input being the left image, and the model output being the right image.

Model: AE, dataset: Illusory Occlusion

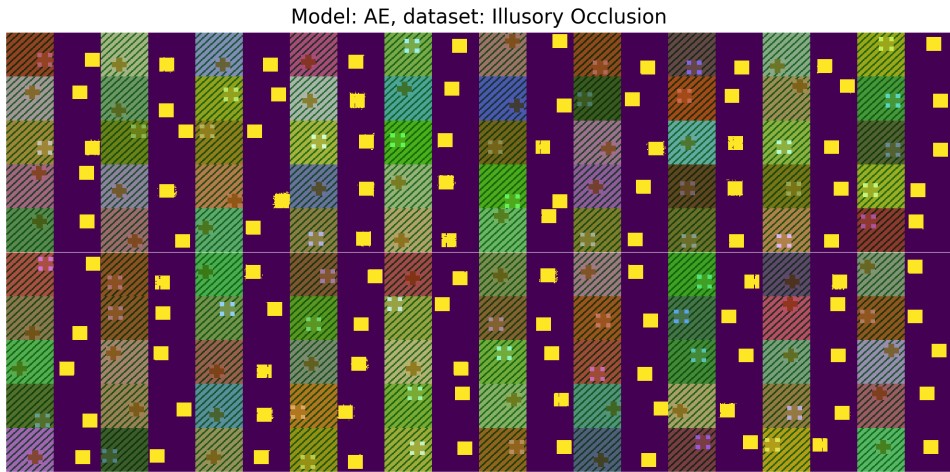

Figure 27: **All Autoencoder outputs for a single seed of the GG Illusory Occlusion dataset.** Inputs and outputs are displayed as horizontal pairs, with the input being the left image, and the model output being the right image.

