# OpenReview forum: "Latent Noise Segmentation: How Neural Noise Leads to the Emergence of Segmentation and Grouping"
_ICLR.cc/2024/Conference — Submitted to ICLR 2024_

### Official Review · Reviewer_cWdJ · 2023-10-31

**Soundness:** 3 good
**Presentation:** 3 good
**Contribution:** 3 good
**Rating:** 6
**Confidence:** 4

**Summary:**

In this manuscript the authors present a method to segment images by adding noise to the latent of a (variational) autoencoder and clustering based on the differences between reconstructions in pixel space. In a couple of controlled experiments on grouping stimuli, this method shows the groupings that are shown by humans.

**Strengths:**

It is an interesting observation that autoencoder objectives alone lead to a representation that separates objects in a similar way as grouping experiments in humans suggest. Also, I think it is an interesting hypothesis that noise at higher levels is translated into correlated noise at lower levels that supports segmentations into objects. The authors present their novel set of gestalt tests that test for the expected grouping results explicitly while being properly image computable.

**Weaknesses:**

I am not convinced by this paper for three reasons:

First, the main evaluation is done on a self made grouping stimulus set, which trains the network quite explicitly to produce the grouping made by human observers as the intended objects are exactly the groups that vary separately in the stimulus generation, by switching color together for example. While this is clearly not direct supervision, it does create a statistical structure that strongly favours the representation of object centred dimensions while there are no variations within objects or over the whole scene. This makes it a lot less surprising that auto encoders find dimensions that create correlations within objects.

Second, the authors emphasise noise strongly in their arguments and go to some length to explain how iid noise might emphasise the local PCs around the stimulus. While I do not think these arguments are technically incorrect, I think they are besides the point. The main step to make this technique work seems to be the shape of the derivative of the decoder around the stimulus. The decoder seems to create correlated changes in the parts that belong to the same object, which yields high similarity for those pixels. It is interesting that this effect is strong enough that few samples are sufficient to separate objects successfully, but in principle any way of estimating this derivative should work. As the noise size here corresponds to the typical delta used to compute the approximate derivative, it is also not surprising that small noise works well. The interesting part is that the derivative creates correlated changes for each object, not that this can be estimated based on noise samples.

Third, I think the connection to human or biological vision is weaker than suggested by the authors. In biological vision we need a segmentation of the internal layer representations, not of the pixels in the image. This requires that the encoder and decoder are in some way related that allows us to connect our noise reconstructions and the encoding elements. Additionally, we would need a biologically plausible clustering algorithm that is based on the found similarities in the noise. Both are not present in the model presented here. Thus, substantial revisions would be necessary to transform the method proposed here into a biologically plausible method for object segmentation.

**Questions:**

Perhaps my questions highlight that this manuscript seems not ideally placed at a machine learning conference:
- I would like to understand how the authors imagine the noise based segmentation to work in biological vision: How does the autoencoder model map to the brain? And what evidence is there that anything like this might actually happen in the brain?
- Does this work in any way for natural scenes? Testing this on existing autoencoders for natural scenes could avoid my concerns about the training data being very targeted to create the patterns observed in humans.
- And what about comparisons to alternative methods? Do other methods for segmentation or grouping get the gestalt tests wrong?

---

> ### Author Response · Authors · 2023-11-18
> **Reply to comments by reviewer cWdj 1/4**
>
> >It is an interesting observation that autoencoder objectives alone lead to a representation that separates objects in a similar way as grouping experiments in humans suggest. Also, I think it is an interesting hypothesis that noise at higher levels is translated into correlated noise at lower levels that supports segmentations into objects.
>
> Thank you for your review and comments. We are happy that you find our work interesting.
>
> >First, the main evaluation is done on a self made grouping stimulus set, which trains the network quite explicitly to produce the grouping made by human observers as the intended objects are exactly the groups that vary separately in the stimulus generation, by switching color together for example. While this is clearly not direct supervision, it does create a statistical structure that strongly favours the representation of object centred dimensions while there are no variations within objects or over the whole scene. This makes it a lot less surprising that auto encoders find dimensions that create correlations within objects.
>
> The motivation behind this choice is a simplification of natural image statistics - the luminance-illuminance-reflectance relationship. For a set of naturalistic images in the same scene, if some surface shares a material, they also share their reflectance, which is the cue of the color in our synthetic dataset. We certainly agree that this relationship does not hold true in all scenarios, but its simplicity lends itself to a more detailed analysis.
>
>
> The cue of color is also not strictly necessary (as seen in CelebA): if there was no cue of color, the information revealed by LNS would allow the remaining covariances within objects, such as their position, to dominate. When viewed in segmentation masks, this would certainly make the segmentation masks appear hollow unless there were other simultaneous cues that affected the entire object at once (e.g., 3D rotation), which we do not have in our simplified 2D datasets. To improve the mathematical rigor of the work, and to make the datasets themselves more accessible, we chose to take the simplification. We believe that investigating the more general case where other cues that are less amenable to mathematical exposition are present is a fruitful future direction, but given the extensive theoretical and empirical analysis we show here, outside the scope of this work.
>
> Finally, we added other model comparisons trained and evaluated on the same datasets to show that even when the models are well-trained (Appendix C2), they fail to exhibit the desired grouping results in our datasets (Table 1; we append this table to another comment you raised, Appendix C1).
>
> >Second, the authors emphasise noise strongly in their arguments and go to some length to explain how iid noise might emphasise the local PCs around the stimulus. While I do not think these arguments are technically incorrect, I think they are besides the point. The main step to make this technique work seems to be the shape of the derivative of the decoder around the stimulus. The decoder seems to create correlated changes in the parts that belong to the same object, which yields high similarity for those pixels. It is interesting that this effect is strong enough that few samples are sufficient to separate objects successfully, but in principle any way of estimating this derivative should work.
>
> We agree with the reviewer that any derivative estimation method should work - the interesting part is that something as naive as noise actually estimates this gradient. Second, we agree that the shape of the derivative of the decoder around the stimulus is the relevant part. To briefly clarify this point: at a high level, the reconstruction-oriented training process optimizing ELBO tends to represent local principle components of the training data samples at the derivative of the output with respect to the latents (for more details, please see Rolinek et al. (2019) and Zietlow et al. (2021)). The distributions of the training samples cause the local principal components to exhibit certain patterns. The derivative's components belonging to one object correlate because the distribution of the training samples does not span in the direction where they do not correlate. Likewise, the derivative's components belonging to different objects are not likely to always correlate since the distributions of the training samples span a space characterizing them as independent. We include a full exposition of this in Appendix A.
>
> To summarize, the distribution of the training samples makes the "shape" of derivatives favorable for our segmentation task (Appendix A). The intriguing fact is that the distribution of the training dataset follows, from the perspective of optics, a simplification of real-world visual scenes, which we discuss in some detail in Appendix E.

---

> ### Author Response · Authors · 2023-11-18
> **Reply to comments by reviewer cWdj 2/4**
>
> > As the noise size here corresponds to the typical delta used to compute the approximate derivative, it is also not surprising that small noise works well. The interesting part is that the derivative creates correlated changes for each object, not that this can be estimated based on noise samples.
>
> The response above should clarify this point - we also have a full exposition of this in Appendix A. Let us know if we can clarify this further.
>
> >Third, I think the connection to human or biological vision is weaker than suggested by the authors. [...] Both are not present in the model presented here. Thus, substantial revisions would be necessary to transform the method proposed here into a biologically plausible method for object segmentation.
>
> Thank you for the comment - the connection to biology was also brought up by another reviewer. Our goal is not to propose an implementation-level plausible model of the visual system, but rather a computational model for perceptual grouping.
>
> To make this clear in the manuscript, we have rephrased parts of the paper:
>
> >_**Abstract**: “Finally, we (4) demonstrate the __ecological plausibility__ of the method by analyzing the sensitivity of the DNN to different magnitudes of noise.”_
>
> => _“Finally, we (4) demonstrate the __practical feasibility__ of the method by analyzing the sensitivity of the DNN to different magnitudes of noise.”_
>
> >_**Abstract**: “Together, our results suggest a novel unsupervised segmentation method requiring few assumptions, a new explanation for the formation of perceptual grouping, __and a potential benefit of neural noise in the visual system__”_
>
> => _“Together, our results suggest a novel unsupervised segmentation method requiring few assumptions, a new explanation for the formation of perceptual grouping, **and a novel potential benefit of neural noise.**”_
>
> >_**Introduction**: “Finally, we study how **ecologically plausible LNS is (that is, how in-principle usable the method is by a biological system)**”_
>
> => _“Finally, we study how **practically feasible LNS is (that is, how in-principle usable the method is by any system constrained by limitations like compute time or noise magnitude, such as the primate visual system)**”_
>
> > _**Introduction**: “Our results suggest that an **ecologically plausible** number of time steps” _
>
> => _“Our results suggest that a **practically feasible** number of time steps”_
>
> > _**Section 3.3**: “To understand how **ecologically plausible** LNS is, we wanted to understand how the amount of noise added in latent space would affect the model’s segmentation performance.”_
>
> => _“To understand how **practically feasible LNS is for a system constrained by compute time or noise magnitude**, we analyzed how the amount of noise added in latent space would affect the model’s segmentation performance.”_
>
> > _**Conclusion**: “With both learning rules, segmentation performance asymptotes quickly, suggesting that the time scale of the segmentation in our models is **ecologically plausible**.”_
>
> => _“With both learning rules, segmentation performance asymptotes quickly, suggesting that the time scale of the segmentation in our models is **practically feasible for constrained systems**.”_
>
> >[...] In biological vision we need a segmentation of the internal layer representations, not of the pixels in the image. This requires that the encoder and decoder are in some way related that allows us to connect our noise reconstructions and the encoding elements. [...]
>
> We would like to clarify on this point that was also alluded to by another reviewer:
> Latent Noise Segmentation is a way to “show” the result of segmentation that the network has already learned during training. Specifically, LNS reveals the shape of the local derivatives around the input sample. This additional information that LNS reveals enables segmentation using a variety of methods. In our paper, we segment by propagating this information through the decoder, and clustering the outputs. Naturally, this segmentation could happen at any downstream layer (thanks to the chain rule: if the clusterability of the representation depends on the shape of the derivatives, which it does, and is present in the last layer, which it is, then it must propagate through the entire decoder and is accessible at any level of representation).
>
> To make this point clearer, we have changed the text in Section 2.2 as follows:
>
> “[...] This means that we can perturb all units simultaneously with independent noise: training on a generic reconstruction loss encourages the network to code in a manner that allows independent noise to reveal the relevant derivative direction, making the representation clusterable for segmentation (Figure1b).
>
> Indeed, the goal of independent noise is not to perform segmentation itself, but to reveal the local neighborhood of the input stimulus and thus enable segmentation, _from the perspective of the decoder_.”

---

> ### Author Response · Authors · 2023-11-18
> **Reply to comments by reviewer cWdj 3/4**
>
> >[...] Additionally, we would need a biologically plausible clustering algorithm that is based on the found similarities in the noise. [...]
>
> Our point is not that Agglomerative Clustering is biologically plausible, but rather that the reason that LNS works is because it makes the representation clusterable. To make this point clearer, we changed Section 2.2 as mentioned above. This was a point raised by other reviewers too, and to dig further into this, we have conducted additional experiments with different clustering algorithms.
> These experiments are in Appendix D1.
>
> | Model      | Kanizsa                 | Closure                 | Continuity              | Proximity               | Gradient Occ.           | Illusory Occ.           |
> |------------|-------------------------|-------------------------|-------------------------|-------------------------|-------------------------|-------------------------|
> | AE+Agg     | 0.859 ± 0.008           | 0.766 ± 0.008           | 0.552 ± 0.008           | **0.996 ± 0.001**       | **0.926 ± 0.009**       | 0.994 ± 0.002           |
> | VAE+Agg    | **0.871 ± 0.005**       | 0.795 ± 0.009           | **0.593 ± 0.012**       | 0.943 ± 0.010           | 0.918 ± 0.002           | 0.974 ± 0.003           |
> | AE+KMeans  | 0.854 ± 0.003           | 0.803 ± 0.001           | 0.553 ± 0.004           | 0.995 ± 0.000           | 0.922 ± 0.006           | **0.998 ± 0.000**       |
> | VAE+KMeans | 0.866 ± 0.003           | **0.852 ± 0.006**       | 0.589 ± 0.006           | 0.921 ± 0.005           | 0.924 ± 0.003           | 0.986 ± 0.001           |
> | AE+Agg'    | 0.557 ± 0.007           | 0.296 ± 0.006           | 0.322 ± 0.004           | 0.929 ± 0.004           | 0.702 ± 0.004           | 0.520 ± 0.027           |
> | VAE+Agg'   | 0.457 ± 0.008           | 0.324 ± 0.004           | 0.302 ± 0.004           | 0.752 ± 0.012           | 0.594 ± 0.009           | 0.438 ± 0.006           |
>
> *Model results for different clustering algorithms (ARI ± standard error of the mean).*
> *Agg = the agglomerative clustering algorithm from `sklearn` with `metric` being "euclidean" and `linkage` being "ward".*
> *Agg' = the agglomerative clustering algorithm from `sklearn` with `metric` being "euclidean" and `linkage` being "complete".*
> *KMeans = the K-means algorithm from `sklearn`.*
> *These scores are obtained for the optimal noise level and number of samples found for Agg.*
>
> In summary, while clustering algorithm can change the results (agglomerative clustering with the Ward linkage does better than Agglomerative Clustering with the Complete linkage), the results do not crucially depend on the clustering algorithm (K-means clustering performs as well as, or better than, Agglomerative Clustering with the Ward linkage).
>
> Furthermore, we now report UMAP visualizations for the clusterability of the examples shown in Figure 4. This additional section can be found in Appendix F2, and further bolsters the claim that the specific clustering method is not important, but rather, that clustering is only a part of the algorithm that allows us to convert already-existing clusters into segmentation masks for visualization.
>
> >Perhaps my questions highlight that this manuscript seems not ideally placed at a machine learning conference
>
> We thank you for the comment, but we respectfully disagree: our goal with this work is to present a computational approach to perceptual grouping using minimal assumptions, to analyze why it works (as you correctly mentioned, noise reveals the shape of the local derivatives), and to analyze different properties about the method. We hope the changes we have made in the manuscript following your suggestions make this clearer.
>
> >I would like to understand how the authors imagine the noise based segmentation to work in biological vision: How does the autoencoder model map to the brain? And what evidence is there that anything like this might actually happen in the brain?
>
> To clarify, our goal is certainly not to propose an implementation-level model of the visual system, but rather, a computational account of perceptual grouping. That being said, our model choice is inspired and motivated by links to biology (VAEs and predictive coding: Marino, 2022; Bouting et al., 2020; VAEs and the Free Energy Principle: Friston, 2010; Mazzaglia, 2022; empirical support for VAEs in macaques: Higgins et al., 2021). While we tried to be as careful as possible when writing the manuscript, we hope that the changes to the text in the manuscript outlined in an earlier reply help clarify this intention.

---

> ### Author Response · Authors · 2023-11-18
> **Reply to comments by reviewer cWdj 4/4**
>
> >Does this work in any way for natural scenes? Testing this on existing autoencoders for natural scenes could avoid my concerns about the training data being very targeted to create the patterns observed in humans. And what about comparisons to alternative methods? Do other methods for segmentation or grouping get the gestalt tests wrong?
>
> To address the concerns about the training data being very targeted, and about how alternative methods do on these datasets, we now report results for two additional SOTA/near-SOTA models: Genesis and Genesis-v2 (Engelcke et al., 2020; 2022).
>
> | Model     | Kanizsa           | Closure           | Continuity        | Proximity         | Gradient Occ.     | Illusory Occ.     |
> |-----------|-------------------|-------------------|-------------------|-------------------|-------------------|-------------------|
> | AE        | 0.859 ± 0.008     | 0.766 ± 0.008     | 0.552 ± 0.008     | **0.996\*** ± 0.001 | 0.926 ± 0.009     | **0.994\*** ± 0.002 |
> | VAE       | **0.871** ± 0.005 | **0.795** ± 0.009 | 0.593 ± 0.012     | 0.943 ± 0.010     | 0.918 ± 0.002     | 0.974 ± 0.003     |
> | AE Rec.   | 0.246 ± 0.006     | 0.064 ± 0.001     | 0.570 ± 0.006     | 0.141 ± 0.001     | **0.982** ± 0.000 | -0.023 ± 0.001    |
> | VAE Rec.  | 0.343 ± 0.004     | 0.084 ± 0.000     | **0.611** ± 0.005 | 0.144 ± 0.001     | 0.977 ± 0.001     | -0.024 ± 0.001    |
> | Genesis   | 0.346             | 0.755             | 0.394             | 0.879             | 0.933             | 0.294             |
> | Genesis-v2| 0.415             | 0.000             | 0.399             | 0.059             | 0.000             | 0.422             |
>
> *Model results (ARI ± standard error of the mean). AE (Autoencoder) and VAE (Variational Autoencoder) model scores reported for the best noise level and number of samples. AE Rec. and VAE Rec. stand for noisy reconstruction control experiments, and (see Section 2.4 for details). Bold indicates top-performing model. For the AE and the VAE, a star (\*) indicates a statistically significant difference in model performance (Bonferroni corrected).*
>
> These results are now reported in the main text in Table 1. While quantitative results for the Genesis model look impressive, a further investigation in Appendix C1 reveals that the decent quantitative performance is driven by good pixel-level predictions in nearly all cases, and not actual Gestalt grouping phenomena (Figure 10). Indeed, the model appears to primarily follow pixel-level information present in the input.
>
> To verify that this result was not caused by a mistake in our implementation, we followed the official implementation of Genesis and Genesis-v2, and show reconstructions on a separate validation set (Appendix C2).
>
> Thank you once again for your review. We believe that the additional clarifications, experiments and analyses address your comments, and we hope that you consider raising your score.

---

> ### Author Response · Authors · 2023-11-18
> **Reply to comments by reviewer cWdj (references)**
>
> References:
>
> _Rolınek, M., Zietlow, D., & Martius, G. (2019). Variational autoencoders pursue pca directions (by accident). IEEE/CVF Conference on Computer Vision and Pattern Recognition (CVPR), pages 12398–12407. doi: 10.1109/CVPR.2019.01269._
>
> _Zietlow, D., Rolinek, M., & Martius, G.. (2021). Demystifying inductive biases for β-vae based architectures. ArXiv, abs/2102.06822. 15_
>
> _Engelcke, M., Kosiorek, A. R., Jones, O. P., & Posner, I. (2020). GENESIS: Generative Scene Inference and Sampling with Object-Centric Latent Representations. In International Conference on Learning Representations (ICLR)._
>
> _Engelcke, M., Jones, O. P., & Posner, I. (2022). GENESIS-V2: Inferring Unordered Object Representations without Iterative Refinement. ArXiv pre-print._
>
> _Boutin, V., Zerroug, A., Jung, M. & Serre, T. (2020). Iterative VAE as a predictive brain model for out-of-distribution generalization. ArXiv pre-print, (arXiv:2012.00557)._
>
> _Marino, J. (2022). Predictive coding, variational autoencoders, and biological connections. Neural Computation, 34(1):1–44._
>
> _Friston, K. (2010). The free-energy principle: a unified brain theory? Nature Reviews Neuroscience, 11(2):127–138. doi: 10.1038/nrn2787._
>
> _Mazzaglia, P., Verbelen, T., Catal, O., & Dhoedt, B. (2022). The free energy principle for perception and action: A deep learning perspective. Entropy, 24(2):301. doi: 10.3390/e24020301._
>
> _Higgins, I., Chang, L., Langston, V., Hassabis, D., Summerfield, C., Tsao, D., & Botvinick, M. (2021). Unsupervised deep learning identifies semantic disentanglement in single inferotemporal face patch neurons. Nature Communications, 12(1):6456. doi: 10.1038/s41467-021-26751-5_

---

> ### Author Response · Authors · 2023-11-22
> **Additional questions?**
>
> Hi,
>
> Thank you again for your feedback. We believe our new analyses, changes to the paper, and comments sufficiently address your concerns and might warrant an improved score. Do you have any additional concerns about the paper?

---

> ### Comment · Reviewer_cWdJ · 2023-11-23
> **Substantial improvements!**
>
> The authors provide a thorough response to my comments and have adjusted their formulations. I have increased my scores accordingly.
> Nonetheless, I am not going to vote strongly in favour of the paper as some of the weaknesses remain. For example, the tasks are still strongly simplified and a closer match or comparability to human vision would still be desirable, even though the authors clarified the situation at least.

---

> > ### Author Response · Authors · 2023-11-23
> > **Thank you**
> >
> > Thank you again for your review and for improving your score following our response!

---

### Official Review · Reviewer_9tYc · 2023-11-01

**Soundness:** 3 good
**Presentation:** 3 good
**Contribution:** 3 good
**Rating:** 8
**Confidence:** 4

**Summary:**

The authors propose a new unsupervised object discovery technique based on variational autoencoders with noise injected to the latent representation. The authors propose an unsupervised object segmentation algorithm that works by computing the difference between reconstructions of an input image from noisy latent representations followed by pixel-wise clustering. The proposed approach is tested on unsupervised perceptual grouping performance on the Good Gestalt (GG) dataset that the authors propose in this submission as well. Quantitative evaluation performed on the GG dataset shows promising evidence of Latent Noise Segmentation discovering objects while being trained on unsupervised image reconstruction. Further analyses on the sensitivity of LNS to latent noise parameters helps better understand the working of VAEs equipped with latent noise segmentation.

**Strengths:**

+ Very intriguing to see emergent perceptual grouping from ANNs trained on unsupervised image reconstruction objectives. Super interesting perspective that latent neuronal noise could lead to learning good gestalt priors.
+ The approach is very straightforward and easy to understand. I think this simplicity is a big strength of the proposed work.
+ Good Gestalt datasets are also a nice addition to the contributions from this work, I hope this benchmark can serve as a good way to measure perceptual grouping abilities of models in the community.
+ I like the additional analyses performed on understanding how latent noise parameters affects emergent grouping.

**Weaknesses:**

- The focus of the paper feels a bit narrow in terms of the architecture / learning objective. Is there something special about VAEs trained with ELBO + LNS that makes them develop emergent grouping, or does LNS generalize across architectural choices and learning rules (say, diffusion-based or adversarial generative models)? Adding a discussion on this would add more value to the submission
- I feel that GG's difficulty could be significantly improved by adding more distractors and/or noise to the background of images. Although the current emergence of grouping looks interesting, I would be even more surprised if the model is learning to discount background noise in its presence, i.e., currently the dataset makes figure-ground organization too simple by providing a largely low-frequency background and I believe GG can be solved merely by using simple rules on low-level feature detectors.
- The authors have covered a variety of unsupervised object discovery approaches such as Slot Attention, Complex-valued autoencoders in related work but have not performed a direct comparison to these baselines in their reported experiments. This makes the paper weaker due to the absence of relevant baselines other than the VAE-based ones currently reported in this version of the paper.

Overall, the strengths of this work marginally outweigh the weaknesses mentioned above, I would be inclined to further improving my score provided the authors convincingly rebut my concerns about this work.

**Questions:**

- Can the authors please comment on whether they experimented with harder versions of GG? Here are a few potential options: (1) Change the size of the square/circle between train and test splits for Kanisza Squares, Closure, Continuity, (2) Use different rotations (without overlap between train and test splits) of the Kanisza squares / Closure squares, (3) Use different colors / textures as backgrounds in all tasks. Any modification of the dataset in the spirit of the modifications suggested here will further strengthen the message that simple low-level statistics don't drive the emergence of perceptual grouping.
- Adding stronger baselines such as the ones the authors have mentioned in related work will help improve my score further.

---

> ### Author Response · Authors · 2023-11-18
> **Reply to comments by reviewer 9tYc 1/3**
>
> >The approach is very straightforward and easy to understand. I think this simplicity is a big strength of the proposed work.
> >
> >Good Gestalt datasets are also a nice addition to the contributions from this work, I hope this benchmark can serve as a good way to measure perceptual grouping abilities of models in the community.
>
> Thank you for the kind comments and for your thorough review. Our hope is indeed that our GG datasets will serve as a relatively low-cost and practical, yet discriminative test of models of perceptual grouping. We believe that our additional baseline results in our revision demonstrate the value of these datasets in discriminating models.
>
> >The focus of the paper feels a bit narrow in terms of the architecture / learning objective. Is there something special about VAEs trained with ELBO + LNS that makes them develop emergent grouping, or does LNS generalize across architectural choices and learning rules (say, diffusion-based or adversarial generative models)? Adding a discussion on this would add more value to the submission
>
> Thank you for the suggestion. We agree this would add value to the paper, and have added a section in Appendix A.1.5 that is referred to in Section 2 of the main text. The section reads:
>
> “In this work, we have focused primarily on the Variational Autoencoder (VAE) in lieu of other generative models, such as Diffusion models (Sohl-Dickstein et al., 2015; Rombach et al., 2022) or Adversarial Generative (Goodfellow et al., 2014; Creswell et al., 2018) models. There are two primary reasons behind this choice: 1) The VAE formulation allows us to neatly follow the argumentation in Zietlow et al. (2021) and Rolınek et al. (2019) to arrive at a more firm understanding of why Latent Noise Segmentation (LNS) works; 2) the mathematical and computational links between the VAE formulation and other biologically plausible coding and learning schemes such as predictive coding (Boutin et al., 2020; Marino, 2022) and the Free Energy Principle (Friston, 2010; Mazzaglia et al., 2022) are clearer than for other generative algorithms to the best of our knowledge.
>
> That being said, the fact that the Autoencoder works relatively well (although qualitatively differently, see Appendix F 2) for segmentation using LNS is interesting, and raises the question of whether other model architectures could also perform well. We suggest that future work could empirically evaluate LNS on other model classes, such as Latent Diffusion models (Rombach et al., 2022) to understand whether the mathematical intuition behind why LNS works also generalizes beyond the simple Autoencoder model class to any model class that learns meaningful representations about images.”
>
> >I feel that GG's difficulty could be significantly improved by adding more distractors and/or noise to the background of images. Although the current emergence of grouping looks interesting, I would be even more surprised if the model is learning to discount background noise in its presence, i.e., currently the dataset makes figure-ground organization too simple by providing a largely low-frequency background and I believe GG can be solved merely by using simple rules on low-level feature detectors.
>
> In a much earlier version of the work, we had simpler versions of some of the GG datasets, which indeed were somewhat too easy. To remedy this, we made the tasks more difficult (e.g., we increased the frequency of the Illusory Occlusion’s background stripes), and added more GG datasets that combine different challenges (e.g., long-range integration of contours in Kanizsa, and inference of objects of the same color in Proximity). The difficulty in our datasets does not come from the individual images themselves per se, but rather from the qualitatively substantial generalization gap between the training set (that depends on simple assumptions) and testing set.
>
> To strengthen our claim about the difficulty of the datasets, we also trained additional baseline models - we give full details in our reply to your next comment.

---

> ### Author Response · Authors · 2023-11-18
> **Reply to comments by reviewer 9tYc 2/3**
>
> >The authors have covered a variety of unsupervised object discovery approaches such as Slot Attention, Complex-valued autoencoders in related work but have not performed a direct comparison to these baselines in their reported experiments. This makes the paper weaker due to the absence of relevant baselines other than the VAE-based ones currently reported in this version of the paper.
>
> Thank you for this suggestion, with which we agree. We now report results for two additional SOTA/near-SOTA models: Genesis and Genesis-v2 (Engelcke et al., 2020; 2022).
>
> | Model     | Kanizsa           | Closure           | Continuity        | Proximity         | Gradient Occ.     | Illusory Occ.     |
> |-----------|-------------------|-------------------|-------------------|-------------------|-------------------|-------------------|
> | AE        | 0.859 ± 0.008     | 0.766 ± 0.008     | 0.552 ± 0.008     | **0.996\*** ± 0.001 | 0.926 ± 0.009     | **0.994\*** ± 0.002 |
> | VAE       | **0.871** ± 0.005 | **0.795** ± 0.009 | 0.593 ± 0.012     | 0.943 ± 0.010     | 0.918 ± 0.002     | 0.974 ± 0.003     |
> | AE Rec.   | 0.246 ± 0.006     | 0.064 ± 0.001     | 0.570 ± 0.006     | 0.141 ± 0.001     | **0.982** ± 0.000 | -0.023 ± 0.001    |
> | VAE Rec.  | 0.343 ± 0.004     | 0.084 ± 0.000     | **0.611** ± 0.005 | 0.144 ± 0.001     | 0.977 ± 0.001     | -0.024 ± 0.001    |
> | Genesis   | 0.346             | 0.755             | 0.394             | 0.879             | 0.933             | 0.294             |
> | Genesis-v2| 0.415             | 0.000             | 0.399             | 0.059             | 0.000             | 0.422             |
>
> *Model results (ARI ± standard error of the mean). AE (Autoencoder) and VAE (Variational Autoencoder) model scores reported for the best noise level and number of samples. AE Rec. and VAE Rec. stand for noisy reconstruction control experiments, and (see Section 2.4 for details). Bold indicates top-performing model. For the AE and the VAE, a star (\*) indicates a statistically significant difference in model performance (Bonferroni corrected).*
>
> These results are now reported in the main text in Table 1. While quantitative results for the Genesis model look impressive, a further investigation in Appendix C1 reveals that the decent quantitative performance is driven by good pixel-level predictions in nearly all cases, and not actual Gestalt grouping phenomena (Figure 10). Indeed, the model appears to primarily follow pixel-level information present in the input.
>
> To verify that this result was not caused by a mistake in our implementation (such as a trivial failure in convergence), we followed the official implementation of Genesis and Genesis-v2, and show reconstructions on a separate validation set (Appendix C2).
>
> >Can the authors please comment on whether they experimented with harder versions of GG? Here are a few potential options:
> (1) Change the size of the square/circle between train and test splits for Kanisza Squares, Closure, Continuity
>
> Although we don’t have quantitative results to report on this, during the design of the GG datasets we did have a variant with different sizes for Illusory Occlusion. Qualitatively, the results did not differ. This is not surprising because Latent Noise Segmentation can be viewed as a form of “simultaneous random latent traversal” (Kingma & Welling, 2013; Higgins et al., 2017). Specifically, in latent traversal you change the activity of one latent unit at a time to understand how that changes the output image. Thus, when adding noise to the latents in the presence of a generative factor of size in the dataset (and thus the latents), size merely provides an additional cue for grouping the contours of the image. A similar effect already exists in the data, induced by the generative factor of position.
>
> We attempted to avoid making the GG datasets difficult for reasons that did not directly contribute to testing for grouping capabilities of the model. This was done to make the datasets, and training on them, as accessible as possible as low-cost tests of specifically grouping. We believe our additional baseline results suggest that we somewhat succeeded in hitting the sweet spot - it is possible to train other models to convergence on the datasets - but together the datasets discriminate between models that perform grouping, and those that do not. Indeed, there is still upwards room on our datasets quantitatively, as even though our approach solves the necessary perceptual grouping conditions, it is clear that models that do so more accurately could improve quantitative scores on these benchmarks further.

---

> ### Author Response · Authors · 2023-11-18
> **Reply to comments by reviewer 9tYc 3/3**
>
> > (2) Use different rotations (without overlap between train and test splits) of the Kanisza squares / Closure squares
>
> The same logic as changing the size applies here - changing a rotational axis corresponds to an additional cue for contour grouping (just like size, but only along a rotational axis instead of a larger-smaller axis).
>
> >  (3) Use different colors / textures as backgrounds in all tasks. Any modification of the dataset in the spirit of the modifications suggested here will further strengthen the message that simple low-level statistics don't drive the emergence of perceptual grouping.
>
> To show that low-level statistics do not indeed drive the emergence of perceptual grouping, we now report two additional SOTA/near-SOTA baseline models that achieve good reconstruction error, and yet fail at the desired perceptual grouping tests even when they get pixel value properties correct (e.g., exact position of elements).
>
> >Adding stronger baselines such as the ones the authors have mentioned in related work will help improve my score further.
>
> Thank you again for the suggestion - we have addressed this above.
>
> >Overall, the strengths of this work marginally outweigh the weaknesses mentioned above, I would be inclined to further improving my score provided the authors convincingly rebut my concerns about this work.
>
> Thank you for your assessment and your review - we believe our additional experiments and clarifications address your concerns and we hope that you consider raising your score.

---

> ### Author Response · Authors · 2023-11-18
> **Reply to comments by reviewer 9tYc (references)**
>
> References:
>
> _Higgins, I., Matthey, L., Pal, A., Burgess, C., Glorot, X., Botvinick, M., Mohamed, S., & Lerchner, A. (2017). beta-VAE: Learning Basic Visual Concepts with a Constrained Variational Framework. International Conference on Learning Representations._
>
> _Kingma, D. P. & Welling, M. (2014). Auto-Encoding Variational Bayes. International Conference on Learning Representations._
>
> _Boutin, V., Zerroug, A., Jung, M. & Serre, T. (2020). Iterative VAE as a predictive brain model for out-of-distribution generalization. ArXiv pre-print, (arXiv:2012.00557)._
>
> _Marino, J. (2022). Predictive coding, variational autoencoders, and biological connections. Neural Computation, 34(1):1–44._
>
> _Friston, K. (2010). The free-energy principle: a unified brain theory? Nature Reviews Neuroscience, 11(2):127–138. doi: 10.1038/nrn2787._
>
> _Mazzaglia, P., Verbelen, T., Catal, O., & Dhoedt, B. (2022). The free energy principle for perception and action: A deep learning perspective. Entropy, 24(2):301. doi: 10.3390/e24020301._
>
> _Engelcke, M., Kosiorek, A. R., Jones, O. P., & Posner, I. (2020). GENESIS: Generative Scene Inference and Sampling with Object-Centric Latent Representations. In International Conference on Learning Representations (ICLR)._
>
> _Engelcke, M., Jones, O. P., & Posner, I. (2022). GENESIS-V2: Inferring Unordered Object Representations without Iterative Refinement. ArXiv pre-print._
>
> _Rombach, R., Blattmann, A., Lorenz, D., Esser, P., & Ommer, B. (2022). High-resolution image synthesis with latent diffusion models. In Proceedings of the IEEE/CVF Conference on Computer Vision and Pattern Recognition (CVPR) (pp. 10684–10695)._
>
> _Sohl-Dickstein, J., Weiss, E., Maheswaranathan, N., & Ganguli, S. (2015). Deep unsupervised learning using nonequilibrium thermodynamics. Proceedings of the 32nd International Conference on Machine Learning (pp. 2256–2265)._
>
> _Goodfellow, I., Pouget-Abadie, J., Mirza, M., Xu, B., Warde-Farley, D., Ozair, S., Courville, A., & Bengio, Y. (2014). Generative adversarial nets. 27._
>
> _Creswell, A., White, T., Dumoulin, V., Arulkumaran, K., Sengupta, B., & Bharath, A. A. (2018). Generative adversarial networks: An overview. IEEE Signal Processing Magazine, 35(1), 53–65. doi: 10.1109/MSP.2017.2765202._

---

> ### Author Response · Authors · 2023-11-22
> **Additional questions?**
>
> Hi,
>
> Thank you again for your feedback. We believe our new analyses, changes to the paper, and comments sufficiently address your concerns and might warrant an improved score. Do you have any additional concerns about the paper?

---

> > ### Comment · Reviewer_9tYc · 2023-11-30
> > **Response to author rebuttal**
> >
> > Dear authors,
> >
> > Thank you very much for your thorough response to my review and other reviews of your submission. I acknowledge that with comparison to the new baselines that have been added in the paper, changes made, results on ablating the clustering method with more choices and detailed rebuttal responses, this paper has become stronger and meets the standards I expect to find at ICLR. I recommend this submission for acceptance, thank you.

---

### Official Review · Reviewer_ipWu · 2023-11-01

**Soundness:** 2 fair
**Presentation:** 3 good
**Contribution:** 1 poor
**Rating:** 3
**Confidence:** 3

**Summary:**

This paper investigates noise in feature space as a means of segmenting images. Several datasets of simple artificial images are developed for this work, each of which is designed to test whether the method exhibits a different Gestalt property. Autoencoders are used to produce latent representations. Small independent noise is repeatedly added to the latent representations. Differences are calculated between pairs of noisy outputs to produce pixel-wise vectors of differences. These vectors are clustered to produce a segmentation. This often results in Gestalt-like segmentations, e.g. segmentation of Kanizsa squares from background.

**Strengths:**

The dataset is a nice contribution, particularly the test set with expected segmentations for images that are expected to elicit various Gestalt phenomena.

The segmentation method is novel (as far as I know) and creative.

**Weaknesses:**

The abstract claims that the results show, “potential benefit of neural noise in the visual system” and the paper repeatedly claims to study the ecologically plausibility of the method. Noise is certainly prevalent in biological vision, but I'm not convinced that there is a substantial connection between this method and biology. Issues include: 1) reliance of the method on small-amplitude noise, whereas spiking noise is closer to Poisson; 2) use of agglomerative clustering; 3) lack of comparison with neural data; 4) need for specialized training datasets to produce Gestalt phenomena; 5) poor performance on natural images.

If the goal is to explain something about biological visual systems, I think much clearer links to biology are needed. It seems to me that this would require substantial changes to the model and/or much more detailed justification of multiple model elements.

To elaborate on point 4 above, the method seems to rely on the design of the training datasets to work properly. For example, to segment Kanizsa squares, an autoencoder is first trained on stimuli that show the squares in a different color than the background color. This kind of dependence is claimed explicitly in the appendix: “If p1 and p2 are pixels belonging to the same object, the way the dataset is generated … dictates that the training samples projected onto the pixel value space will only stretch in a direction where there exists a strict linear ratio between the values of the two objects.” Humans don't require such specialized training to experience Gestalt perception.

To elaborate on point 5 above, the method is not meant to be practical, but its low practicality is also a concern for biological plausibility. It is not extensively tested on natural or otherwise practical images but in addition to the Gestalt dataset images, it was tested on celebrity faces and the paper claims that the method “often finds a semantically meaningful segmentation of face-hair-background”. However, examination of the results (Appendix A.3) shows that the results are generally poor. According to Table 1, even in these artificial circumstances, the model only outperforms the control in 4/6 cases. The control is the same model with noise applied to the output rather than to the latent representation. It could be interesting to also contrast with established strong segmentation methods, particularly if this method agreed with humans in conditions where others don’t. However, some promise of strong performance would be needed in an convincing model of human vision.

**Questions:**

What size are the GG datasets?

---

> ### Author Response · Authors · 2023-11-18
> **Reply to comments by reviewer ipWu 1/4**
>
> > The dataset is a nice contribution, particularly the test set with expected segmentations for images that are expected to elicit various Gestalt phenomena.
> >
> >The segmentation method is novel (as far as I know) and creative.
>
> Thank you for your detailed review. We are glad you found our datasets interesting, and approach novel and creative. We have made several improvements to our manuscript as a result of your comments, and believe it now more carefully represents the primary goal of our work: that of a computational approach to perceptual grouping and segmentation, rather than an implementation-level explanation of specific neural processes in the primate visual system.
>
> >The abstract claims that the results show, “potential benefit of neural noise in the visual system” and the paper repeatedly claims to study the ecologically plausibility of the method. Noise is certainly prevalent in biological vision, but I'm not convinced that there is a substantial connection between this method and biology.
>
> We attempted to be extraordinarily careful with our phrasing in the paper, not making any direct claims about human/primate vision. Indeed, in the discussion, we explicitly state:
> _“Finally, while we do not make any claims about the primate visual system, our results suggest a potential benefit of independent noise in the visual system.“_
>
> That being said, we apologize that there remains a lack of clarity about the objectives of the work in its framing. In the work, we study our approach of perceptual grouping and segmentation from a computational perspective, looking for a computational solution (rather than an implementation-level solution) to the well-known Gestalt principles of grouping. Our main goal is not to provide an exhaustive explanation of segmentation in primates, but rather to take a substantial and theoretically-grounded step towards an explanation of the Gestalt grouping laws in a computational framework. Our work is certainly inspired by biology, and our model choice was specifically driven by their computational and experimental links to biology (_Predictive coding: Marino, 2022; Bouting et al., 2020; Free Energy Principle: Friston, 2010; Mazzaglia, 2022; empirical support in macaques: Higgins et al., 2021_), but we agree that this does not constitute a link to biology.
>
> As such, we have rephrased our terminology in parts of the paper, specifically removing mentions of ecological plausibility:
>
> >_**Abstract**: “Finally, we (4) demonstrate the __ecological plausibility__ of the method by analyzing the sensitivity of the DNN to different magnitudes of noise.”_
>
> => _“Finally, we (4) demonstrate the __practical feasibility__ of the method by analyzing the sensitivity of the DNN to different magnitudes of noise.”_
>
> >_**Abstract**: “Together, our results suggest a novel unsupervised segmentation method requiring few assumptions, a new explanation for the formation of perceptual grouping, __and a potential benefit of neural noise in the visual system__”_
>
> => _“Together, our results suggest a novel unsupervised segmentation method requiring few assumptions, a new explanation for the formation of perceptual grouping, **and a novel potential benefit of neural noise.**”_
>
> >_**Introduction**: “Finally, we study how **ecologically plausible LNS is (that is, how in-principle usable the method is by a biological system)**”_
>
> => _“Finally, we study how **practically feasible LNS is (that is, how in-principle usable the method is by any system constrained by limitations like compute time or noise magnitude, such as the primate visual system)**”_
>
> > _**Introduction**: “Our results suggest that an **ecologically plausible** number of time steps” _
>
> => _“Our results suggest that a **practically feasible** number of time steps”_
>
> > _**Section 3.3**: “To understand how **ecologically plausible** LNS is, we wanted to understand how the amount of noise added in latent space would affect the model’s segmentation performance.”_
>
> => _“To understand how **practically feasible LNS is for a system constrained by compute time or noise magnitude**, we analyzed how the amount of noise added in latent space would affect the model’s segmentation performance.”_
>
> > _**Conclusion**: “With both learning rules, segmentation performance asymptotes quickly, suggesting that the time scale of the segmentation in our models is **ecologically plausible**.”_
>
> => _“With both learning rules, segmentation performance asymptotes quickly, suggesting that the time scale of the segmentation in our models is **practically feasible for constrained systems**.”_

---

> ### Author Response · Authors · 2023-11-18
> **Reply to comments by reviewer ipWu 2/4**
>
> >  Issues include: 1) reliance of the method on small-amplitude noise, whereas spiking noise is closer to Poisson
>
> Our analyses show that for Latent Noise Segmentation, the noise scale is only meaningful in a comparative, not an absolute sense. What we believe matters is the ratio of noise scale to latent traversal scale (as mentioned by reviewer zkrR). As such, without knowing the scale of the true signal vs the scale of independent noise, a direct implementation-level comparison (especially given how much we do not know about neural coding and noise in cortex) seems difficult in principle. Indeed, we show that for the AE model, various different levels of noise result in good performance, and that if we fix our model coding scheme (in the case of the VAE, where we fix the latent prior to be normally distributed), we consistently find a small range of optimal noise values (Figures 5 and 6).
>
> >Issues include: 2) use of agglomerative clustering
>
> We have made changes in the manuscript to further clarify that the point of LNS is not that we apply a specific clustering algorithm, but that the noise added in latent space reveals a clusterable representation - one that was not clusterable prior to adding noise:
>
> Section 2.2: “[...] This means that we can perturb all units simultaneously with independent noise: training on a generic reconstruction loss encourages the network to code in a manner that allows independent noise to reveal the relevant derivative direction, making the representation clusterable for segmentation (Figure1b).
> Indeed, the goal of independent noise is not to perform segmentation itself, but to reveal the local neighborhood of the input stimulus and thus enable segmentation, _from the perspective of the decoder_.”
>
> To further demonstrate that the specific clustering algorithm is not critical to the method, we have added a new set of experiments in Appendix D1.
>
> | Model      | Kanizsa                 | Closure                 | Continuity              | Proximity               | Gradient Occ.           | Illusory Occ.           |
> |------------|-------------------------|-------------------------|-------------------------|-------------------------|-------------------------|-------------------------|
> | AE+Agg     | 0.859 ± 0.008           | 0.766 ± 0.008           | 0.552 ± 0.008           | **0.996 ± 0.001**       | **0.926 ± 0.009**       | 0.994 ± 0.002           |
> | VAE+Agg    | **0.871 ± 0.005**       | 0.795 ± 0.009           | **0.593 ± 0.012**       | 0.943 ± 0.010           | 0.918 ± 0.002           | 0.974 ± 0.003           |
> | AE+KMeans  | 0.854 ± 0.003           | 0.803 ± 0.001           | 0.553 ± 0.004           | 0.995 ± 0.000           | 0.922 ± 0.006           | **0.998 ± 0.000**       |
> | VAE+KMeans | 0.866 ± 0.003           | **0.852 ± 0.006**       | 0.589 ± 0.006           | 0.921 ± 0.005           | 0.924 ± 0.003           | 0.986 ± 0.001           |
> | AE+Agg'    | 0.557 ± 0.007           | 0.296 ± 0.006           | 0.322 ± 0.004           | 0.929 ± 0.004           | 0.702 ± 0.004           | 0.520 ± 0.027           |
> | VAE+Agg'   | 0.457 ± 0.008           | 0.324 ± 0.004           | 0.302 ± 0.004           | 0.752 ± 0.012           | 0.594 ± 0.009           | 0.438 ± 0.006           |
>
> *Table 2: Model results for different clustering algorithms (ARI ± standard error of the mean).*
> *Agg = the agglomerative clustering algorithm from `sklearn` with `metric` being "euclidean" and `linkage` being "ward".*
> *Agg' = the agglomerative clustering algorithm from `sklearn` with `metric` being "euclidean" and `linkage` being "complete".*
> *KMeans = the K-means algorithm from `sklearn`.*
> *These scores are obtained for the optimal noise level and number of samples found for Agg.*
>
> In summary, while clustering algorithm can change the results (agglomerative clustering with the Ward linkage does better than Agglomerative Clustering with the Complete linkage), the results do not crucially depend on the clustering algorithm (K-means clustering performs as well as, or better than, Agglomerative Clustering with the Ward linkage).
>
> In addition, we now report UMAP visualizations for the clusterability of the examples shown in Figure 4. This additional section can be found in Appendix F2, and further bolsters the claim that the specific clustering method is not important, but rather, that clustering is only a part of the algorithm that allows us to convert already-existing clusters into segmentation masks for visualization.

---

> ### Author Response · Authors · 2023-11-18
> **Reply to comments by reviewer ipWu 3/4**
>
> > Issues include: 3) lack of comparison with neural data
>
> While we agree that comparing to neural data would be a nice addition, we wanted to focus on a behavioral result in combination with an in-depth theoretical account of the phenomenon. A substantial reason for this is a recent movement in the community of modeling human vision (Bowers et al., 2022) that argues that predicting neural activity has gained a large amount of traction in the literature, while targeted modeling of specific phenomena in human vision with controlled experiments has fallen by the wayside.
>
> Here, we aim to explicitly address the latter: we take some of the most classical results in human vision with over a century of history, and model them directly with control experiments, and provide a detailed exposition of why and how our methodology works. Specifically, because of the relative simplicity of our approach we are able to formulate links between basic image statistics, and segmentation using our method (illuminance-luminance-reflectance, Appendix A1 & E1). We thus strongly believe this is a timely contribution from _also_ this perspective (in addition to the contribution of the methodology itself), even if it slightly deviates from the mainstream of neural predictivity analysis.
>
> > Issues include: 4) need for specialized training datasets to produce Gestalt phenomena; 5) poor performance on natural images
>
> We reply to these in detail below.
>
> >If the goal is to explain something about biological visual systems, I think much clearer links to biology are needed. It seems to me that this would require substantial changes to the model and/or much more detailed justification of multiple model elements.
>
> Our goal here is indeed __not__ to explain something about biological visual systems directly, but to provide a detailed computational account of perceptual grouping and segmentation using minimal assumptions with a deep learning model that has computational links to primate vision, and some empirical support (_Predictive coding: Marino, 2022; Bouting et al., 2020; Free Energy Principle: Friston, 2010; Mazzaglia, 2022; empirical support in macaques: Higgins et al., 2021_). We believe that the changes we have made to the manuscript in response to your first question also emphasize this point further.
>
> >To elaborate on point 4 above, the method seems to rely on the design of the training datasets to work properly. For example, to segment Kanizsa squares, an autoencoder is first trained on stimuli that show the squares in a different color than the background color. This kind of dependence is claimed explicitly in the appendix: “If p1 and p2 are pixels belonging to the same object, the way the dataset is generated … dictates that the training samples projected onto the pixel value space will only stretch in a direction where there exists a strict linear ratio between the values of the two objects.” Humans don't require such specialized training to experience Gestalt perception.
>
> We certainly agree that humans do not train on the GG datasets, but the GG datasets capture a basic relationship that is true in naturalistic scenes: the illuminance-luminance-reflectance relationship. Specifically, for any objects in the naturalistic world made up of the same material, their material guarantees a shared reflectance of the object in a given lighting condition, fulfilling the basic assumption we make about the GG datasets. We certainly agree that this relationship does not hold true in all scenarios, but its simplicity lends itself to a more detailed analysis.
>
> Furthermore, removing the cue of color would still yield information meaningful for segmentation, but would require fundamentally different processing method and theoretical understanding to leverage them. If there was no cue of color, the information revealed by LNS would allow the sets of squares to be separated, but the segmentation masks for Proximity would reveal segmentation masks that are hollow (rather than entirely painted in). This is because the relevant cue now separating the objects would be only that of position, and thus, when propagating the information revealed by LNS all the way to the pixel level, only the edges of the squares would contain information about objecthood. As such, an additional mechanism of e.g. spreading activity might be useful (e.g., Linsley et al., 2018 comes to mind as a relevant architectural consideration) for forming full segmentation masks. Investigating this direction, however, we believe is outside of the scope of the current work given the extensive theoretical and empirical analysis we show about the simplified case that relies on color.
>
> To further strengthen our claim that the GG datasets are meaningfully difficult tasks for model falsification, we ran additional baseline models (discussed further in the response to your next question).

---

> ### Author Response · Authors · 2023-11-18
> **Reply to comments by reviewer ipWu 4/4**
>
> >To elaborate on point 5 above, the method is not meant to be practical, but its low practicality is also a concern for biological plausibility. It is not extensively tested on natural or otherwise practical images but in addition to the Gestalt dataset images, it was tested on celebrity faces and the paper claims that the method “often finds a semantically meaningful segmentation of face-hair-background”. However, examination of the results (Appendix A.3) shows that the results are generally poor. According to Table 1, even in these artificial circumstances, the model only outperforms the control in 4/6 cases. The control is the same model with noise applied to the output rather than to the latent representation. It could be interesting to also contrast with established strong segmentation methods, particularly if this method agreed with humans in conditions where others don’t. However, some promise of strong performance would be needed in an convincing model of human vision.
>
> Thank you for the suggestion - we think your suggestion of evaluating other models on our datasets is relevant and important. We now report results for two additional SOTA/near-SOTA models: Genesis and Genesis-v2 (Engelcke et al., 2020; 2022).
>
> | Model     | Kanizsa           | Closure           | Continuity        | Proximity         | Gradient Occ.     | Illusory Occ.     |
> |-----------|-------------------|-------------------|-------------------|-------------------|-------------------|-------------------|
> | AE        | 0.859 ± 0.008     | 0.766 ± 0.008     | 0.552 ± 0.008     | **0.996\*** ± 0.001 | 0.926 ± 0.009     | **0.994\*** ± 0.002 |
> | VAE       | **0.871** ± 0.005 | **0.795** ± 0.009 | 0.593 ± 0.012     | 0.943 ± 0.010     | 0.918 ± 0.002     | 0.974 ± 0.003     |
> | AE Rec.   | 0.246 ± 0.006     | 0.064 ± 0.001     | 0.570 ± 0.006     | 0.141 ± 0.001     | **0.982** ± 0.000 | -0.023 ± 0.001    |
> | VAE Rec.  | 0.343 ± 0.004     | 0.084 ± 0.000     | **0.611** ± 0.005 | 0.144 ± 0.001     | 0.977 ± 0.001     | -0.024 ± 0.001    |
> | Genesis   | 0.346             | 0.755             | 0.394             | 0.879             | 0.933             | 0.294             |
> | Genesis-v2| 0.415             | 0.000             | 0.399             | 0.059             | 0.000             | 0.422             |
>
> *Model results (ARI ± standard error of the mean). AE (Autoencoder) and VAE (Variational Autoencoder) model scores reported for the best noise level and number of samples. AE Rec. and VAE Rec. stand for noisy reconstruction control experiments, and (see Section 2.4 for details). Bold indicates top-performing model. For the AE and the VAE, a star (\*) indicates a statistically significant difference in model performance (Bonferroni corrected).*
>
> These results are now reported in the main text in Table 1. While quantitative results for the Genesis model look impressive, a further investigation in Appendix C1 reveals that the decent quantitative performance is driven by good pixel-level predictions in nearly all cases, and not actual Gestalt grouping phenomena (Figure 10). Indeed, the model appears to primarily follow pixel-level information present in the input.
>
> To verify that this result was not caused by a mistake in our implementation, we followed the official implementation of Genesis and Genesis-v2, and show reconstructions on a separate validation set (Appendix C2).
>
> Finally, we would like to emphasize out that coping with full-scale naturalistic images and revealing computational principles of perceptual grouping are fundamentally different objectives of research. Here, we mainly focused on the latter.
>
> > What size are the GG datasets?
>
> Training: 30,000 images, validation: 300 images that are not present in the training set, test: 100 images. We have also added this information in Appendix E1.
>
> Thank you again for your comments, which have helped us improve our manuscript. We believe our substantial additional experiments and analyses, combined with a careful rephrasing of critical parts in the paper address your comments, and we hope that you consider increasing your score as a result of these changes.

---

> ### Author Response · Authors · 2023-11-18
> **Reply to comments by reviewer ipWu (references)**
>
> References:
>
> _Boutin, V., Zerroug, A., Jung, M. & Serre, T. (2020). Iterative VAE as a predictive brain model for out-of-distribution generalization. ArXiv pre-print, (arXiv:2012.00557)._
>
> _Marino, J. (2022). Predictive coding, variational autoencoders, and biological connections. Neural Computation, 34(1):1–44._
>
> _Friston, K. (2010). The free-energy principle: a unified brain theory? Nature Reviews Neuroscience, 11(2):127–138. doi: 10.1038/nrn2787._
>
> _Mazzaglia, P., Verbelen, T., Catal, O., & Dhoedt, B. (2022). The free energy principle for perception and action: A deep learning perspective. Entropy, 24(2):301. doi: 10.3390/e24020301._
>
> _Higgins, I., Chang, L., Langston, V., Hassabis, D., Summerfield, C., Tsao, D., & Botvinick, M. (2021). Unsupervised deep learning identifies semantic disentanglement in single inferotemporal face patch neurons. Nature Communications, 12(1):6456. doi: 10.1038/s41467-021-26751-5_
>
> _Bowers, J. S., Malhotra, G., Dujmovic, M., Montero, M. L., Tsvetkov, C., Biscione, V., Puebla, G., Adolfi, F., Hummel, J. E., Heaton, R. F., Evans, B. D., Mitchell, J., & Blything, R. (2022). Deep Problems with Neural Network Models of Human Vision. Behavioral and Brain Sciences, pages 1–74. Doi: 10.1017/S0140525X22002813._
>
> _Linsley, D., Kim, J., Veerabadran, V., Windolf, C., & Serre, T. (2018). Learning long-range spatial dependencies with horizontal gated recurrent units. Advances in Neural Information Processing Systems._
>
> _Engelcke, M., Kosiorek, A. R., Jones, O. P., & Posner, I. (2020). GENESIS: Generative Scene Inference and Sampling with Object-Centric Latent Representations. In International Conference on Learning Representations (ICLR)._
>
> _Engelcke, M., Jones, O. P., & Posner, I. (2022). GENESIS-V2: Inferring Unordered Object Representations without Iterative Refinement. ArXiv pre-print._

---

> ### Author Response · Authors · 2023-11-22
> **Additional questions?**
>
> Hi,
>
> Thank you again for your feedback. We believe our new analyses, changes to the paper, and comments sufficiently address your concerns and might warrant an improved score. Do you have any additional concerns about the paper?

---

### Official Review · Reviewer_zkrR · 2023-11-03

**Soundness:** 4 excellent
**Presentation:** 4 excellent
**Contribution:** 4 excellent
**Rating:** 8
**Confidence:** 5

**Summary:**

The authors demonstrate that segmentation and grouping features emerge unsupervisedly by injecting iid noise in the latent space of VAE and AE. They show this by designing a simple algorithm that compute relative differences of reconstructed images from latent code corrupted by noise on top of which they stack et simple clustering algorithm. In addition, the author have built a large dataset consisting of different grouping/segmentation tasks corresponding to various Gestalt aspects of perception such as closure, continuity, proximity, etc.

**Strengths:**

- well-grounded in the vision science field, enough references
- the idea is well motivated by the search of a role for neural noise and tested in artificial neural network
- the performances of the proposed method are extensively tested on a relevant dataset that is build for this purpose (that one counts twice)
- comparison of VAE and AE is provided together with a control of the idea of adding noise in the latent space
- amount of added noise and step required in the algorithm are also evaluated

**Weaknesses:**

**Minor weaknesses**
- The role of the post-processing step is not evaluated : does agglomerative clustering play a big role ? There are other standard clustering methods that could be tested.
- Even if it's not designed for the segmentation of natural images and if it's likely to not perform very well compared to SOTA algorithms it is worth testing it. I have in mind a recent paper (Vacher et al 2022) in which deep neural network features are evaluated for segmenting natural images.
- Latent space of VAE are known to enable appealing morphing between natural images (by linearly interpolating the latent code) so I am wondering what would be the segmentation related uncertainty that could be obtained with this method ... I guess this would require a more involved post-processing step.
- Other neural network architectures are not tested (GANs, Normalizing flows, ...)

**Minor remark**
- In table 1, bold should be used for every best performing model, eg also for Continuity and Gradient Occ.


Refs :
- Vacher, J., Launay, C., & Coen-Cagli, R. (2022). Flexibly regularized mixture models and application to image segmentation. Neural Networks, 149, 107-123.


**Post-response update**
I thank the authors for their response and considering it together with the other reviews, I am increasing my score so that it is now a clear accept.

**Questions:**

see above

---

> ### Author Response · Authors · 2023-11-18
> **Reply to comments by reviewer zkrR 1/3**
>
> > __Soundness:__ 4 excellent
> >
> > __Presentation:__ 4 excellent
> >
> >__Contribution:__ 4 excellent
>
> Thank you for your review, and your positive assessment of our paper.
>
> >The role of the post-processing step is not evaluated : does agglomerative clustering play a big role ? There are other standard clustering methods that could be tested.
>
> We have conducted additional experiments to evaluate how critical the specific clustering algorithm is for our results. We provide results below and in Appendix D1.
>
> | Model      | Kanizsa                 | Closure                 | Continuity              | Proximity               | Gradient Occ.           | Illusory Occ.           |
> |------------|-------------------------|-------------------------|-------------------------|-------------------------|-------------------------|-------------------------|
> | AE+Agg     | 0.859 ± 0.008           | 0.766 ± 0.008           | 0.552 ± 0.008           | **0.996 ± 0.001**       | **0.926 ± 0.009**       | 0.994 ± 0.002           |
> | VAE+Agg    | **0.871 ± 0.005**       | 0.795 ± 0.009           | **0.593 ± 0.012**       | 0.943 ± 0.010           | 0.918 ± 0.002           | 0.974 ± 0.003           |
> | AE+KMeans  | 0.854 ± 0.003           | 0.803 ± 0.001           | 0.553 ± 0.004           | 0.995 ± 0.000           | 0.922 ± 0.006           | **0.998 ± 0.000**       |
> | VAE+KMeans | 0.866 ± 0.003           | **0.852 ± 0.006**       | 0.589 ± 0.006           | 0.921 ± 0.005           | 0.924 ± 0.003           | 0.986 ± 0.001           |
> | AE+Agg'    | 0.557 ± 0.007           | 0.296 ± 0.006           | 0.322 ± 0.004           | 0.929 ± 0.004           | 0.702 ± 0.004           | 0.520 ± 0.027           |
> | VAE+Agg'   | 0.457 ± 0.008           | 0.324 ± 0.004           | 0.302 ± 0.004           | 0.752 ± 0.012           | 0.594 ± 0.009           | 0.438 ± 0.006           |
>
> *Table 2: Model results for different clustering algorithms (ARI ± standard error of the mean).*
> *Agg = the agglomerative clustering algorithm from `sklearn` with `metric` being "euclidean" and `linkage` being "ward".*
> *Agg' = the agglomerative clustering algorithm from `sklearn` with `metric` being "euclidean" and `linkage` being "complete".*
> *KMeans = the K-means algorithm from `sklearn`.*
> *These scores are obtained for the optimal noise level and number of samples found for Agg.*
>
> In summary, while clustering algorithm can change the results (agglomerative clustering with the Ward linkage does better than Agglomerative Clustering with the Complete linkage), the results do not crucially depend on the clustering algorithm (K-means clustering performs as well as, or better than, Agglomerative Clustering with the Ward linkage).
>
> Furthermore, to clarify that the reason LNS works is not because of the clustering algorithm but because it makes the underlying representation clusterable, we have changed the text in Section 2.2:
>
> “[...] This means that we can perturb all units simultaneously with independent noise: training on a generic reconstruction loss encourages the network to code in a manner that allows independent noise to reveal the relevant derivative direction, making the representation clusterable for segmentation (Figure1b).
> Indeed, the goal of independent noise is not to perform segmentation itself, but to reveal the local neighborhood of the input stimulus and thus enable segmentation, _from the perspective of the decoder_.”
>
> In addition, we now report UMAP visualizations for the clusterability of the examples shown in Figure 4. This additional section can be found in Appendix F2.

---

> ### Author Response · Authors · 2023-11-18
> **Reply to comments by reviewer zkrR 2/3**
>
> >Even if it's not designed for the segmentation of natural images and if it's likely to not perform very well compared to SOTA algorithms it is worth testing it. I have in mind a recent paper (Vacher et al 2022) in which deep neural network features are evaluated for segmenting natural images.
>
> We believe that dealing with naturalistic images and understanding the perceptual biases that lead to grouping are different directions of research and chose to focus on the latter in this work. To address a more naturalistic case where real image statistics play a role, we evaluated our models on the CelebA dataset (Appendix B). We think your suggestion is interesting and would have loved to do it, but we also believe that training generative models on meaningfully large datasets is quite an engineering challenge due to the changes in scale and architecture required (e.g., Child, 2021; Vahdat & Kautz, 2020).
>
> Instead, as suggested by other reviewers, to contextualize our model in the extant literature, we have instead conducted control experiments on other SOTA/near-SOTA models on our datasets (Genesis and Genesis-v2; Engelcke et al, 2020; 2022), results for which we now report in the main text, Table 1.
>
> | Model     | Kanizsa           | Closure           | Continuity        | Proximity         | Gradient Occ.     | Illusory Occ.     |
> |-----------|-------------------|-------------------|-------------------|-------------------|-------------------|-------------------|
> | AE        | 0.859 ± 0.008     | 0.766 ± 0.008     | 0.552 ± 0.008     | **0.996\*** ± 0.001 | 0.926 ± 0.009     | **0.994\*** ± 0.002 |
> | VAE       | **0.871** ± 0.005 | **0.795** ± 0.009 | 0.593 ± 0.012     | 0.943 ± 0.010     | 0.918 ± 0.002     | 0.974 ± 0.003     |
> | AE Rec.   | 0.246 ± 0.006     | 0.064 ± 0.001     | 0.570 ± 0.006     | 0.141 ± 0.001     | **0.982** ± 0.000 | -0.023 ± 0.001    |
> | VAE Rec.  | 0.343 ± 0.004     | 0.084 ± 0.000     | **0.611** ± 0.005 | 0.144 ± 0.001     | 0.977 ± 0.001     | -0.024 ± 0.001    |
> | Genesis   | 0.346             | 0.755             | 0.394             | 0.879             | 0.933             | 0.294             |
> | Genesis-v2| 0.415             | 0.000             | 0.399             | 0.059             | 0.000             | 0.422             |
>
> *Model results (ARI ± standard error of the mean). AE (Autoencoder) and VAE (Variational Autoencoder) model scores reported for the best noise level and number of samples. AE Rec. and VAE Rec. stand for noisy reconstruction control experiments, and (see Section 2.4 for details). Bold indicates top-performing model. For the AE and the VAE, a star (\*) indicates a statistically significant difference in model performance (Bonferroni corrected).*
>
> While quantitative results for the Genesis model look impressive, a further investigation in Appendix C1 reveals that the decent quantitative performance is driven by good pixel value predictions in nearly all cases, and not actual Gestalt grouping phenomena (Figure 10). Indeed, the model appears to primarily follow pixel value information present in the input.
>
> To verify that this result was not caused by a mistake in our implementation (e.g. total failure of model training), we followed the official implementation of Genesis and Genesis-v2, and show reconstructions on a separate validation set (Appendix C2).
>
> >Latent space of VAE are known to enable appealing morphing between natural images (by linearly interpolating the latent code) so I am wondering what would be the segmentation related uncertainty that could be obtained with this method ... I guess this would require a more involved post-processing step.
>
> This is quite an interesting question and something we only briefly mention in the paper (Section 2.2, first paragraph). We believe that Latent Noise Segmentation can be thought of as simultaneous random latent traversal of all the latent nodes. As such, Figure 6a may help answer your question: we know that in VAEs, the prior is set to $N(0, 1)$, and as such have a meaningful reference point for the “size” of the latent traversal. What is interesting is that in the case of the VAE, performance decreases substantially at higher levels of noise (i.e., larger “random latent traversals”). This suggests that the level of noise (magnitude of the “random latent traversal”) must be large enough to get a reliable signal, but small enough to not meaningfully cause uncertainty about the objects in the input image.

---

> ### Author Response · Authors · 2023-11-18
> **Reply to comments by reviewer zkrR 3/3**
>
> > Other neural network architectures are not tested (GANs, Normalizing flows, ...)
>
> While we think this would be nice to do, we have in this work specifically taken the approach of understanding in detail (both empirically and theoretically) the underpinnings of Latent Noise Segmentation as a method that can, in a principled way, solve outstanding problems in machine perception. To support this argument, as mentioned above, we have conducted additional experiments on other extant well-known models to bolster our claim about the difficulty of the datasets, and about how LNS uniquely solves the datasets we present in the paper.
>
> >In table 1, bold should be used for every best performing model, eg also for Continuity and Gradient Occ.
>
> Thank you for pointing this out - we have fixed it.
>
> Thank you once again for your thorough review which has helped improve our manuscript. We believe that our additional experiments address your questions and we hope that you consider increasing your score.

---

> ### Author Response · Authors · 2023-11-18
> **Reply to comments by reviewer zkrR (references)**
>
> References:
>
> _Child, R. (2021). Very Deep VAEs Generalize Autoregressive Models and Can Outperform Them on Images. In International Conference on Learning Representations (ICLR)._
>
> _Vahdat, A. & Kautz, J. (2020). NVAE: A Deep Hierarchical Variational Autoencoder. Advances in Neural Information Processing Systems._
>
> _Engelcke, M., Kosiorek, A. R., Jones, O. P., & Posner, I. (2020). GENESIS: Generative Scene Inference and Sampling with Object-Centric Latent Representations. In International Conference on Learning Representations (ICLR)._
>
> _Engelcke, M., Jones, O. P., & Posner, I. (2022). GENESIS-V2: Inferring Unordered Object Representations without Iterative Refinement. ArXiv pre-print._

---

> ### Author Response · Authors · 2023-11-22
> **Thank you**
>
> Thank you again for your review and for updating your score following our response!

---

### Official Review · Reviewer_DELR · 2023-11-04

**Soundness:** 3 good
**Presentation:** 3 good
**Contribution:** 3 good
**Rating:** 8
**Confidence:** 3

**Summary:**

This paper studies the role of neural noise in the formation of perceptual groups. They show how one can obtain segmentation maps from a (V)AE simply through the injection of noise in the latent space, without any supervision on a segmentation task. Concretely, N noisy versions of a latent vector are passed through the decoder to result in N slightly different outputs. The difference maps between two consecutive outputs are then turned into one segmentation map using a clustering algorithm. The paper posits that the reason this process reveals perceptual groups in a scene is because pixels belonging to the same group tend to co-vary. Indeed, an experiment using a novel dataset, Good Gestalt, reveals cognitively viable segmentation maps that seem to obey the Gestalt laws. The appendix also includes an experiment on natural images (CelebA). In the paper's conclusion, neural noise is put forward as a potential mechanism for perceptual grouping.

**Strengths:**

Asking how perceptual grouping may occur without explicit supervision is important considering how many modern models rely on such supervision, whereas our own visual system arguably handles it differently. Moreover, I think the papers takes an interesting and fresh take on it by studying how noise might in fact be beneficial to visually separate objects.

The extent of the analyses (mathematical accounts, extra results on CelebA, noise sensitivity analysis) etc. is impressive. More than once I wrote down something I intended to inquire about, only to see that exact question already addressed a bit further down the paper.

It's a well-prepared manuscript, written with care.

**Weaknesses:**

The potential weaknesses I have spotted could very well rather be unclarities, so I'll save it for the "Questions" section.

**Questions:**

1)
I'm unclear on the extent to which Latent Noise Segmentation is a method to reveal the perceptual groups already formed inside the network through some mechanism or another, versus a mechanism that gives rise to perceptual groups in its own right. Is the hypothesis that LNS is a way of "doing" segmentation or is it a way to "show" the result of segmentation, if that makes sense?

2)
The training samples in the GG dataset often combine two Gestalt cues. For example, in Proximity, the parts closest to each other are also similar in color. Was that crucial to the results? Would the segmentations maps no longer obey the law of Proximity if there was no color cue during training?

In Section 2, does pretraining refer to training on GG before doing the noise injection, or was there any pretraining on natural images? If so, it would be interesting to see segmentation maps for GG test images before training on GG train images. Would it group by continuity just by learning from natural statistics, for example?

I think it might be worthwhile to show examples of VAE outputs without noise (i.e., the actual reconstructions, not the segmentation maps). If it groups the pixels of the Kanizsa squares together supposedly "perceives" the square, does it output its illusionary contours?

3)
"Time steps" is used to refer to the number of noisy samples needed for a segmentation map. Does it bear any relation with actual time in the human visual system?

4)
The paper refers to ecological validity and biological plausibility, but I'd love to see a little more elaboration on how exactly a biological visual system would potentially carry out the operations suggested here (e.g., how can we picture the 'clustering' being done)?

---

> ### Author Response · Authors · 2023-11-18
> **Reply to comments by reviewer DELR 1/3**
>
> > The extent of the analyses (mathematical accounts, extra results on CelebA, noise sensitivity analysis) etc. is impressive. More than once I wrote down something I intended to inquire about, only to see that exact question already addressed a bit further down the paper.
> It's a well-prepared manuscript, written with care.
>
> Thank you for your thorough review and this kind comment - we are glad to hear you enjoyed the extent of the analyses and exposition in our paper.
>
> >I'm unclear on the extent to which Latent Noise Segmentation is a method to reveal the perceptual groups already formed inside the network through some mechanism or another, versus a mechanism that gives rise to perceptual groups in its own right. Is the hypothesis that LNS is a way of "doing" segmentation or is it a way to "show" the result of segmentation, if that makes sense?
>
> In this phrasing, LNS is a way to “show” the result of segmentation that the network has already learned during training. Specifically, LNS reveals the shape of the local derivatives around the input sample. This additional information that LNS reveals enables segmentation using a variety of methods. In our paper, we segment by propagating this information through the decoder, and clustering the outputs.
>
> To make this point clearer, we have changed the text in Section 2.2 as follows:
> “[...] This means that we can perturb all units simultaneously with independent noise: training on a generic reconstruction loss encourages the network to code in a manner that allows independent noise to reveal the relevant derivative direction, making the representation clusterable for segmentation (Figure 1b).
> Indeed, the goal of independent noise is not to perform segmentation itself, but to reveal the local neighborhood of the input stimulus and thus enable segmentation, _from the perspective of the decoder_.”
>
>
> >The training samples in the GG dataset often combine two Gestalt cues. For example, in Proximity, the parts closest to each other are also similar in color. Was that crucial to the results? Would the segmentations maps no longer obey the law of Proximity if there was no color cue during training?
>
> Color dependence in this specific context is crucial for forming a full segmentation mask, but it is not crucial for object separation.
>
> We do not necessarily see this as a bad thing for three reasons:
> 1. The basic assumption is justified by optics: the luminance-illuminance-reflectance relationship of different materials. Nearby objects that consist of the same materials have the same reflectance, providing a cue for grouping in naturalistic images.
> 2. This basic assumption which holds for many naturalistic scenes is also a key part of our theoretical intuition linking image statistics and segmentation in Appendices A1 and E1. Keeping the cue of color, while not crucial for object separation, makes the theoretical analysis substantially simpler and clearer.
> 3. Removing the cue of color would still yield information meaningful for segmentation, but would require a different processing method and theoretical understanding to leverage them: if there was no cue of color, the information revealed by LNS would allow the sets of squares to be separated, but the segmentation masks for Proximity would reveal segmentation masks that are hollow (rather than entirely painted in). This is because the relevant cue now separating the objects would be only that of position, and thus, when propagating the information revealed by LNS all the way to the pixel level, only the edges of the squares would contain information about objecthood. As such, an additional mechanism of e.g. spreading activations for forming full segmentation masks from detected edges might be useful (e.g., Linsley et al., 2018 comes to mind as a relevant architectural consideration). Investigating this direction, however, we believe is outside of the scope of the current work given the extensive theoretical and empirical analysis we show about the simplified case that relies on color. Furthermore, in more complex datasets, there are other object-level cues (such as rotation and parallax) that can be utilized in lieu of only color.

---

> ### Author Response · Authors · 2023-11-18
> **Reply to comments by reviewer DELR 2/3**
>
> > In Section 2, does pretraining refer to training on GG before doing the noise injection, or was there any pretraining on natural images? If so, it would be interesting to see segmentation maps for GG test images before training on GG train images. Would it group by continuity just by learning from natural statistics, for example?
>
> Pretraining refers to normal AE/VAE training on any desired dataset of interest before a noise injection. We did not use any naturalistic images in pre-training, and contained our training and analysis only to the GG and CelebA datasets. That being said, we agree that it would be interesting to see whether naturalistic image statistics would, in practice, result in good segmentation performance on our datasets. Unfortunately, training generative models on large datasets is a substantial and costly engineering effort (see e.g., Child, 2021; Vahdat & Kautz, 2020), and we hoped to demonstrate the scalability of the approach using the CelebA dataset. Due to the substantially different architecture of other architectures and the limited size of naturalistic datasets for generative models, we believe that exploring LNS in faithfully naturalistic pre-trained models would be a meaningfully large independent project - here, we have focused solely on the empirical and theoretical underpinnings of the novel method.
>
> > I think it might be worthwhile to show examples of VAE outputs without noise (i.e., the actual reconstructions, not the segmentation maps). If it groups the pixels of the Kanizsa squares together supposedly "perceives" the square, does it  output its illusionary contours?
>
> Thank you for the suggestion. We now plot the model reconstructions in Appendix F3, and report:
>
> “To support our control experiment, we also visualize the model output reconstructions (Figure 14). In accordance with the control experiments, we find that the outputs in some cases contain some information about the identity of the objects, even when it does not veridically exist in the input image. However, as reported in Table 1, the amount of information in the reconstructions is not as high as it is when using Latent Noise Segmentation. This is because of the tension between reconstructing an image well (which leads to poor segmentation without LNS in test samples), and segmenting the image components based on their color values. In other words, the fact that the model is still able to segment in some cases can be understood as an artefact caused by the small size of our model and imperfect reconstruction performance. The training process implicitly optimizes toward the success of LNS segmentation, but against the success of direct reconstruction segmentation.”
>
> > "Time steps" is used to refer to the number of noisy samples needed for a segmentation map. Does it bear any relation with actual time in the human visual system?
>
> While one could interpret it that way, there is a clear degree of separation in our model and the human visual system (leaving aside the more philosophical considerations): our models deal with continuous activation values, while neurons spike. While it is possible to model deep neural networks as spiking networks (for example, by matching the activation values of a continuous network with the firing rates of spiking networks; Rueckauer et al., 2017), we have not considered it here to get an estimate of how the time steps should be implemented in the corresponding implementation of spiking networks.

---

> ### Author Response · Authors · 2023-11-18
> **Reply to comments by reviewer DELR 3/3**
>
> > The paper refers to ecological validity and biological plausibility, but I'd love to see a little more elaboration on how exactly a biological visual system would potentially carry out the operations suggested here (e.g., how can we picture the 'clustering' being done)?
>
> We want to be very careful with what we claim in the paper in terms of ecological validity, and indeed have removed mentions to that as suggested by other reviewers. However, we still think this is an interesting question. Another reviewer brought up noise sampling and neuronal spikes, which we think is an interesting example. In our deep networks, we are sampling continuous values with noise, while the neurons in the brain observe spikes from upstream neurons. Even in spiking networks, a plausibly fast sampling method may not be impossible, as previous work suggests that spike distributions may automatically estimate probability distributions (Ma et al., 2006), which could be one way in which this additive noise could be sampled efficiently.
>
> Neural implementation of clustering (binding) is also currently an open research question. One such example is the temporal correlation hypothesis (Singer & Gray, 1995) - however, LNS is not strictly about the specific implementation of individual components, but about the clusterability of the representation. We have correspondingly edited the main text as mentioned above (Section 2.2). Our focus is on a computational account of perceptual grouping that attempts to rely on minimal assumptions.
>
> We thank you again for your review, which has helped improve our manuscript. We believe these additional experiments, analyses, and writing changes address your comments, and we hope that you consider raising your score or confidence in your score.

---

> ### Author Response · Authors · 2023-11-18
> **Reply to comments by reviewer DELR (references)**
>
> References:
>
> _Linsley, D., Kim, J., Veerabadran, V., Windolf, C., & Serre, T. (2018). Learning long-range spatial dependencies with horizontal gated recurrent units. Advances in Neural Information Processing Systems._
>
> _Child, R. (2021). Very Deep VAEs Generalize Autoregressive Models and Can Outperform Them on Images. ICLR._
>
> _Vahdat, A. & Kautz, J. (2020). NVAE: A Deep Hierarchical Variational Autoencoder. Advances in Neural Information Processing Systems._
>
> _Ma, W., Beck, J., Latham, P. & Pouget, A. (2006). Bayesian inference with probabilistic population codes. Nature Neuroscience._
>
> _Rueckauer, B., Lungu, I., Hu, Y., Pfeiffer, M., Liu, S. (2017). Conversion of Continuous-Valued Deep Networks to Efficient Event-Driven Networks for Image Classification. Frontiers in Neuroscience._
>
> _Singer, W. & Gray, C. M. (1995). Visual feature integration and the temporal correlation hypothesis. Annu Rev Neurosci. 18:555-86. doi: 10.1146/annurev.ne.18.030195.003011._

---

> > ### Comment · Reviewer_DELR · 2023-11-22
> > **Thank you**
> >
> > I'd like to thank the authors for their time and effort in addressing my questions in such detail. Overall, my questions have been answered satisfactorily. I appreciate the extra figure showing the reconstructions, along with the interpretation of those results. While I can sympathize with some of the concerns raised by other reviewers, I can also see that the authors have made significant improvements to their original manuscript (more careful description of the scope/purpose, extra comparisons to prior literature, results on the importance of the specific clustering algorithm) in response. All of these elements combined, reaffirm my initial positive assessment of the paper. I think the work is interesting and makes for a valuable contribution.

---

> > > ### Author Response · Authors · 2023-11-23
> > > **Thank you**
> > >
> > > Thank you again for your review and for reaffirming your positive assessment of our paper following our response!

---

### Author Response · Authors · 2023-11-18
**Overview response to all reviewers 1/2**

We would like to thank all the reviewers for their very valuable comments, which helped us substantially improve the manuscript. We have made revisions to the manuscript with several clarifications, additional experiments and analyses, and we believe that the changes we have made to the paper as a result of the comments have made the paper clearer and more sound.

We were delighted that the reviewers found many positive things to say about the manuscript. Many reviewers were happy with the novelty and creativity of the method, pointing out the simplicity of our method as a key strength of our work. Reviewers were generally happy with the depth of our analysis, which we have further improved in our revision. Most reviewers were also happy with the presentation of our work, and we have taken steps to address any remaining concerns regarding it in our revision. Finally, reviewers were interested in our Good Gestalt datasets, which they found well-founded in the extant literature. We will make the datasets available upon publication, and we believe they will serve as a good platform for testing the perceptual grouping capabilities of models in the future.

There were three comments on weaknesses that multiple reviewers pointed out, that focused around additional experiments to contextualize our model with extant literature, additional control experiments of the clustering algorithm, and our phrasing of connections to biology in the paper. We outline responses to these comments here, and we also reply to each of the reviewers’ comments individually.

1. **Lack of comparisons to other models.**

We agree with the reviewers that the paper lacked baseline results that contextualize both how good Latent Noise Segmentation is, as well as how difficult the Good Gestalt datasets are as tests of perceptual grouping.

We ran two additional SOTA/near-SOTA models, (Genesis and Genesis-v2; Engelcke et al, 2020; 2022), which we now report in Table 1:

| Model     | Kanizsa           | Closure           | Continuity        | Proximity         | Gradient Occ.     | Illusory Occ.     |
|-----------|-------------------|-------------------|-------------------|-------------------|-------------------|-------------------|
| AE        | 0.859 ± 0.008     | 0.766 ± 0.008     | 0.552 ± 0.008     | **0.996\*** ± 0.001 | 0.926 ± 0.009     | **0.994\*** ± 0.002 |
| VAE       | **0.871** ± 0.005 | **0.795** ± 0.009 | 0.593 ± 0.012     | 0.943 ± 0.010     | 0.918 ± 0.002     | 0.974 ± 0.003     |
| AE Rec.   | 0.246 ± 0.006     | 0.064 ± 0.001     | 0.570 ± 0.006     | 0.141 ± 0.001     | **0.982** ± 0.000 | -0.023 ± 0.001    |
| VAE Rec.  | 0.343 ± 0.004     | 0.084 ± 0.000     | **0.611** ± 0.005 | 0.144 ± 0.001     | 0.977 ± 0.001     | -0.024 ± 0.001    |
| Genesis   | 0.346             | 0.755             | 0.394             | 0.879             | 0.933             | 0.294             |
| Genesis-v2| 0.415             | 0.000             | 0.399             | 0.059             | 0.000             | 0.422             |

*Model results (ARI ± standard error of the mean). AE (Autoencoder) and VAE (Variational Autoencoder) model scores reported for the best noise level and number of samples. AE Rec. and VAE Rec. stand for noisy reconstruction control experiments, and (see Section 2.4 for details). Bold indicates top-performing model. For the AE and the VAE, a star (\*) indicates a statistically significant difference in model performance (Bonferroni corrected).*

In summary, we found that the baseline models are outperformed by ours, and even when they succeed quantitatively, reaching comparable performance with our models, they fail to capture the desired perceptual grouping effect (Appendix C1). To verify that the baseline model failure is not due to a trivial error (such as the model failing to train on the datasets), we used the official implementation of the models, and verify reconstruction quality in Appendix C2.

---

> ### Author Response · Authors · 2023-11-18
> **Overview response to all reviewers 2/2**
>
> 2. **Importance of the clustering algorithm.**
>
> Several reviewers wondered whether the specific clustering algorithm we use is an important consideration in our work. To address this, we now report model performance using a total of three clustering methods:
>
> 1. Agglomerative Clustering with Ward linkage (our original result)
> 2. Agglomerative Clustering with Complete linkage
> 3. K-means clustering
>
>
> | Model      | Kanizsa                 | Closure                 | Continuity              | Proximity               | Gradient Occ.           | Illusory Occ.           |
> |------------|-------------------------|-------------------------|-------------------------|-------------------------|-------------------------|-------------------------|
> | AE+Agg     | 0.859 ± 0.008           | 0.766 ± 0.008           | 0.552 ± 0.008           | **0.996 ± 0.001**       | **0.926 ± 0.009**       | 0.994 ± 0.002           |
> | VAE+Agg    | **0.871 ± 0.005**       | 0.795 ± 0.009           | **0.593 ± 0.012**       | 0.943 ± 0.010           | 0.918 ± 0.002           | 0.974 ± 0.003           |
> | AE+KMeans  | 0.854 ± 0.003           | 0.803 ± 0.001           | 0.553 ± 0.004           | 0.995 ± 0.000           | 0.922 ± 0.006           | **0.998 ± 0.000**       |
> | VAE+KMeans | 0.866 ± 0.003           | **0.852 ± 0.006**       | 0.589 ± 0.006           | 0.921 ± 0.005           | 0.924 ± 0.003           | 0.986 ± 0.001           |
> | AE+Agg'    | 0.557 ± 0.007           | 0.296 ± 0.006           | 0.322 ± 0.004           | 0.929 ± 0.004           | 0.702 ± 0.004           | 0.520 ± 0.027           |
> | VAE+Agg'   | 0.457 ± 0.008           | 0.324 ± 0.004           | 0.302 ± 0.004           | 0.752 ± 0.012           | 0.594 ± 0.009           | 0.438 ± 0.006           |
>
> *Model results for different clustering algorithms (ARI ± standard error of the mean).*
> *Agg = the agglomerative clustering algorithm from `sklearn` with `metric` being "euclidean" and `linkage` being "ward".*
> *Agg' = the agglomerative clustering algorithm from `sklearn` with `metric` being "euclidean" and `linkage` being "complete".*
> *KMeans = the K-means algorithm from `sklearn`.*
> *These scores are obtained for the optimal noise level and number of samples found for Agg.*
>
> We find that while the specific choice of clustering algorithm may change performance (Agglomerative Clustering with Ward linkage outperforms Complete linkage), it is not a critical design decision: K-means clustering also performs equally well, or better.
> We further clarify in Section 2.2 that the reason Latent Noise Segmentation works is not because of the particular clustering method, but because noise reveals the local derivative neighborhood of the input stimulus, forming a clusterable representation that largely agrees with the general pattern of human perception.
>
> 3. **Clarifications about links to biology, and phrasing of the paper in general.**
>
> While we attempted to be careful in our phrasing in our initial submission, we appreciate the reviewers noting that we were not careful enough. We have clarified the scope of the work in multiple places, including the abstract and conclusion. We make no claims about a specific implementation of Latent Noise Segmentation in the primate visual system, but rather seek to provide an in-depth computational account of perceptual grouping with the use of independent noise. We believe that this is in and of itself a timely and meaningful contribution.
>
> &nbsp;
>
> We once again thank all reviewers for their valuable comments which we believe we have addressed, and hope that any reviewers consider raising their scores as a result.
>
> &nbsp;
>
> References:
>
> _Engelcke, M., Kosiorek, A. R., Jones, O. P., & Posner, I. (2020). GENESIS: Generative Scene Inference and Sampling with Object-Centric Latent Representations. In International Conference on Learning Representations (ICLR)._
>
> _Engelcke, M., Jones, O. P., & Posner, I. (2022). GENESIS-V2: Inferring Unordered Object Representations without Iterative Refinement. ArXiv pre-print._

---

### Author Response · Authors · 2023-11-23
**Discussion period summary**

We thank all reviewers for their reviews, and for recognizing the **simplicity and originality** of our work (“_the segmentation method is novel (as far as I know) and creative_” -ipWu, “_I think this simplicity is a big strength of the proposed work_” – 9tYc, “_the authors present their novel set of gestalt tests_” – cWdj, “_I think the paper takes an interesting and fresh take_” – DELR), the **rigor** of the work (“_the extent of the analyses etc. is impressive_” – DELR, “_the performances of the proposed method are extensively tested on a relevant dataset_” – zkrR, “_I like the additional analyses performed on understanding how latent noise parameters affects emergent grouping_” – 9tYc), and our **presentation** (“_well-grounded in the vision science field_” – zkrR, “_it’s a well-prepared manuscript, written with care_” – DELR).

During the discussion period, the reviewers brought up three main concerns, which we addressed both in an overview comment, as well as in individual replies to all the reviewers:

1.	**Lack of comparisons to other models.** We evaluated additional models in our revision, which are now included in the paper. Our method overall outperforms these additional baseline models both quantitatively and qualitatively.

2.	**Reliance on the specific clustering algorithm.** We ran additional experiments using two other clustering algorithms, and found that the specific clustering algorithm is not critical to the results in the paper, consistent with our discussion. Moreover, we point out that the core of our method is ability to extract a clusterable representation using noise. To clarify this, we modified our text and added UMAP visualizations.

3.	**Clarifications about links to biology, and phrasing of the paper in general.** We made additional clarifications and amendments to the paper’s phrasing, which made our paper’s contribution clearer. Specifically, that our contribution is not an implementation-level account of biological visual segmentation, but rather the discovery of a computational principle of segmentation that aligns with well-documented human phenomena.

Reviewers responded positively to our changes:

-	Reviewer DELR stated that their “questions have been answered satisfactorily.” They reaffirmed their initial positive assessment of the paper (score: **8**, confidence: 3).

-	Reviewer zkrR responded to our revisions by increasing their score from **6 to 8** (confidence: 5).

-	Reviewer cWdj felt that a closer match or comparability to human vision would still be desirable, but felt that we clarified the situation, and that our changes constituted “substantial improvements.” As a result, they increased their score from **3 to 6** (confidence: 4).

-	While we felt that we addressed the comments of reviewers 9tYc (score: **6**, confidence: 4) and ipWu (score: **3**, confidence: 3) with additional experiments and modifications to the text, they have not responded to our revisions or comments, or modified their original reviews as of the writing of this summary.

We thank the reviewers again for their insightful comments which substantially improved the paper and we are happy that taken together, the reviewers feel that our paper is a novel, creative, and sound contribution.

---

### Meta-Review · Area_Chair_PWVb · 2023-12-11

**Metareview:**

The paper presents an interesting hypothesis regarding the role of noise in perceptual organization. Reviewers praised the approach for its simplicity and originality and the quality of the analyses. But, there were also significant weaknesses brought up during the review process.

A major weakness of the paper, discussed between the reviewers, was that the model did not do well on natural images. Even for the synthetic/toy dataset proposed, the model had to be trained and tested on individual datasets. While one of the reviewers felt that the paper was strong enough despite this limitation, the other reviewers agreed that this was a critical limitation of the approach. The AC agrees that it is a very significant limitation because the synthetic datasets are simple and may not approximate the challenge associated with perceptual organization well enough. A convincing demonstration would involve a model trained on natural images, which is then shown to exhibit sensitivity to Gestalt-like properties when tested on artificial stimuli.

Additional weaknesses included a lack of comparison to stronger baselines and the use of hierarchical clustering to derive segmentation masks. Still, both of these criticisms appeared to be addressed well during the rebuttal. Another point of discussion between the reviewers and the authors included a lack of sufficient connection to biology (acknowledged by the authors during the rebuttal). While this was seen as a limitation, there was general agreement that this was not a crucial limitation of the paper.

While there was general agreement that the paper improved during the reviewing process, ultimately, the experiments as they stand failed to convince a majority of reviewers and the AC.

For all these reasons, the AC recommends the paper be rejected.

**Justification For Why Not Higher Score:**

While the paper proposes an interesting hypothesis regarding the role of noise in perceptual organization, the model is always trained and tested on individual synthetic/toy "good gestalt" datasets which casts doubts regarding the strength of the claims.

**Justification For Why Not Lower Score:**

NA

---

### Decision · Program_Chairs · 2024-01-16

Reject